# Pre-trained Speech Processing Models Contain Human-Like Biases that Propagate to Speech Emotion Recognition

**Isaac Slaughter[1], Craig Greenberg[2], Reva Schwartz[2], Aylin Caliskan[1]**
[1]University of Washington, [2]NIST
{is28,aylin}@uw.edu,{craig.greenberg,reva.schwartz}@nist.gov

## Abstract

Previous work has established that a person's demographics and speech style affect how well speech processing models perform for them. But where does this bias come from? In this work, we present the Speech Embedding Association Test (SpEAT), a method for detecting bias in one type of model used for many speech tasks: pre-trained models. The SpEAT is inspired by word embedding association tests in natural language processing, which quantify intrinsic bias in a model's representations of different concepts, such as race or valence—something's pleasantness or unpleasantness—and capture the extent to which a model trained on large-scale socio-cultural data has learned human-like biases. Using the SpEAT, we test for six types of bias in 16 English speech models (including 4 models also trained on multilingual data), which come from the wav2vec 2.0, HuBERT, WavLM, and Whisper model families. We find that 14 or more models reveal positive valence (pleasantness) associations with abled people over disabled people, with European-Americans over African-Americans, with females over males, with U.S. accented speakers over non-U.S. accented speakers, and with younger people over older people. Beyond establishing that pre-trained speech models contain these biases, we also show that they can have real world effects. We compare biases found in pre-trained models to biases in downstream models adapted to the task of Speech Emotion Recognition (SER) and find that in 66 of the 96 tests performed (69%), the group that is more associated with positive valence as indicated by the SpEAT also tends to be predicted as speaking with higher valence by the downstream model. Our work provides evidence that, like text and image-based models, pre-trained speech based-models frequently learn human-like biases when trained on large-scale socio-cultural datasets. Our work also shows that bias found in pre-trained models can propagate to the downstream task of SER.

## 1 Introduction

Recent approaches to many speech tasks rely on pre-trained models: large models trained to learn speech representations (multi-dimensional matrices referred to as embeddings) from large-scale corpora, which can be adapted to a variety of tasks (Feng and Chaspari, 2020; Niu et al., 2020; Yang et al., 2021). In computer vision and natural language processing, methods called Embedding Association Tests (EATs) have been used to evaluate biases in the ways pre-trained models represent social groups, allowing researchers to identify bias early in the machine learning pipeline, before it propagates to downstream tasks (Wolfe et al., 2023; Wolfe and Caliskan, 2022a,b; Steed and Caliskan, 2021; Guo and Caliskan, 2021; Toney-Wails and Caliskan, 2021; Caliskan et al., 2016). In this paper, we present the first intrinsic association and bias evaluation method for pre-trained models in speech processing: the Speech Embedding Association Test (SpEAT).

A SpEAT measures bias related to two social groups in a speech model, and produces an effect size $d$, which when positive indicates that the speech model favors the social group that humans also tend to favor. We evaluate the SpEAT first by testing whether it reveals positive effect sizes, (congruent with human stereotypes), for biases related to twelve social groups. The types of bias that we study have been documented in large populations, and all involve associations with valence, a term often used in psychology literature to describe emotions. Valence is frequently equated with "pleasantness" or "pleasure," where positive valence indicates something pleasant and negative valence indicates something unpleasant (Russell, 1980; Morgan, 2019; Nielen et al., 2009). We study associations with valence due to its role as a primary dimension of affect, strong signal in speech and language, and determinant of how people form

attitudes (Barrett, 2006; Sharot and Garrett, 2016). Using the SpEAT, we find that 15 of 16 models we test show bias for U.S. accented speakers over non-U.S. accented speakers (where relative to non-U.S. accented speakers, U.S. accented speakers are more associated with positive valence than negative valence ), 15 of 16 models show bias for young speakers over old speakers, 14 of 16 models show bias for female speakers over male speakers, 14 of 16 models show bias for European-American speakers over African-American speakers, and 14 of 16 models show bias for abled speakers over disabled speakers. Our results indicate that, as with models trained in other modalities, models trained on large corpora of speech data also learn human-like biases.

To understand the potential impact of these biases, we evaluate whether results found with the SpEAT are indicative of downstream effects. We do so by considering the task of Speech Emotion Recognition (SER), which predicts emotions based on speech (Mohammad, 2022). We compare SpEAT scores for two social groups to disparities in a downstream model's predictions of valence for speech from the groups.[1] We find that social groups that have positive associations with valence in pre-trained models also tend to be predicted by downstream SER models as more positive in valence: For 66 of 96 (69%) SpEATs performed, a downstream SER model tends to predict speech from the social group favored in the SpEAT as more positively valenced than speech from the other group considered.

As negative valence is associated with anger and sadness (Morgan, 2019), this could potentially result in speech from one group of people being translated as more frequently angry than speech from another in a speech translation system that considered emotion when performing translations, or speech from one group being treated as more frequently sad than speech from another in a diagnostic model used in a mental health setting, two suggested applications for SER models (El Ayadi et al., 2011). Beyond showing that biases found in pre-trained models can propagate, our work adds to the growing body of evidence showing bias in

Automated Emotion Recognition (AER), one of many issues that have been raised concerning this area of research (Mohammad, 2022).

Finally, we provide an approach for studying how the number of stimuli (sample size) used in an EAT can change its results. EATs performed in other modalities have used fixed numbers of stimuli to represent social groups and concepts when measuring bias. To evaluate how an EAT score changes when differing sample sizes are used, we calculate bootstrap estimates of the Standard Error (SE) at different sample sizes (Hesterberg, 2011). The SE measures how much a statistic would change if it were calculated repeatedly based on new data, and lower SE values indicate that there is less uncertainty associated with the statistic. We find that the SE of the SpEAT decreases sharply as the number of stimuli used to represent social groups increases, for example that increasing the number of stimuli from 2 to 10 can lead to an increase in precision by a factor of two. Our results show that the number of stimuli used in an EAT can have a large effect on the uncertainty of its effect size $d$.[2]

## 2 Background and Related Work

We now survey related work on pre-trained models in speech processing, as well as work measuring bias in speech models and models in other modalities.

**Pre-Trained Models in Speech Processing** Pre-trained models initially learn from large quantities of unstructured data, for example corpora containing hundreds of thousands of hours of speech, and can then be adapted using smaller structured datasets to a variety of specific tasks (Bommasani et al., 2021). Pre-trained models used for speech processing often train on speech samples that are of variable lengths (Liu et al., 2022a). After initial pre-training, a downstream model for a specific task may then be created based on the pre-trained model, using transfer learning. This is often done using fine-tuning, whereby slight architectural changes are made to the model before it is retrained on a smaller dataset, or feature extraction, which treats the pre-trained model as a pre-processing step for a downstream task-specific model (Peters et al., 2019). Pre-trained models are often transformer based, and with feature extraction a user extracts numerical representations of input data, (also called

---

[1]Previous work studying propagation between intrinsic bias in models and bias in how they perform downstream has suggested that upstream bias is more likely to be indicative of downstream bias if the biases are conceptually similar (Steed et al., 2022; Goldfarb-Tarrant et al., 2021). Because the SpEAT relates to associations with valence, we choose SER as a related downstream task.

[2]Code for this work is available at https://github.com/isaacOnline/SpEAT

embeddings), after each layer in the neural network. The user then employs these embeddings as input data when training a downstream model for a new task (Yang et al., 2021; Peters et al., 2019).

**Bias in Speech Models** Previous research measuring bias in speech processing models largely studies differences in performance on specific speech tasks, for data sourced from people of differing social groups. Social group-based performance comparisons exist for Automated Speech Recognition (ASR) (Tatman, 2017; Tatman and Kasten, 2017; Koenecke et al., 2020; Feng et al., 2021; Liu et al., 2022b; Riviere et al., 2021), Speaker Verification or Speaker Identification (SID) (Hutiri and Ding, 2022; Fenu et al., 2021, 2020; Fenu and Marras, 2022; Chen et al., 2022b; Meng et al., 2022), as well as a number of other speech tasks (Meng et al., 2022; Hutiri et al., 2023). Differences in model performance based on the gender (Tatman, 2017; Tatman and Kasten, 2017; Chen et al., 2022b; Feng et al., 2021; Liu et al., 2022b; Hutiri and Ding, 2022; Fenu et al., 2020, 2021; Fenu and Marras, 2022; Riviere et al., 2021), dialect (Tatman, 2017; Tatman and Kasten, 2017), race (Koenecke et al., 2020; Tatman and Kasten, 2017; Chen et al., 2022b; Riviere et al., 2021), age (Fenu et al., 2020, 2021), city (Koenecke et al., 2020), nationality (Hutiri and Ding, 2022), and native language (Feng et al., 2021) of the speaker have been tested. While these works have established that some speech systems perform worse for people of different social groups (depending on the system and group), pre-trained models have only been studied by testing for performance differences across speech styles in downstream models, as by Meng et al. (2022).

**Embedding Association Tests** EATs are methods for quantifying bias in representation learning models, first introduced by Caliskan et al. (2016, 2017). EATs were originally adapted from Implicit Association Tests (IATs), extensively validated tests used for indirectly measuring implicit associations and bias in humans (Greenwald et al., 1998, 2022). EATs measure the association of two sets of target concepts (e.g., European American and African American) with two sets of attributes (e.g., positively valenced and negatively valenced stimuli). EATs often use positive and negative valence as attributes, due to valence's role in belief formation. EATs were originally used for measuring biases in word embeddings, but have since been used to study biases in sentence encoders (May et al.,

2019), image encoders (Steed and Caliskan, 2021), and multimodal vision-language models (Wolfe and Caliskan, 2022a). These tests have shown that large-scale socio-cultural data, such as text and images, can be a source of implicit associations and biases.

## 3 Methodology

Building on EATs, the SpEAT measures how a model's embeddings of stimuli representing two *target concepts*, (for example female and male), relate to its embeddings of stimuli representing two *attribute concepts* (for example positive and negative valence). After extracting embeddings for stimuli corresponding to the four different concepts from a speech model, the SpEAT is carried out by measuring the relative cosine similarities between the speech models' embeddings of samples representing the four concepts. Let $X$ and $Y$ be the sets of embeddings representing the target concepts, for example embeddings derived from speech samples from female ($X$) and male ($Y$) speakers respectively, and $A$ and $B$ be the set of embeddings representing the attribute concepts, such as embeddings derived from speech samples rated as being positive ($A$) and negative ($B$) in valence, respectively. Similar to other EATs, we present an effect size metric, the SpEAT $d$, which shows which of the target sets, $X$ or $Y$, is relatively more similar to the first attribute set, $A$, than to the second, $B$. As EATs were originally based on IATs, the SpEAT $d$ is based on the IAT $D$, an adaptation of Cohen's $d$ that uses unified standard deviation rather than pooled standard deviation (Greenwald et al., 2003). Values of Cohen's $d$ of 0.20, 0.50 and 0.80 were originally introduced as being small, medium, and large (Cohen, 1977).

To calculate the SpEAT $d$, as for past EATs, first the mean cosine similarity between a target embedding $w$ and each embedding $a \in A$ is calculated, followed by the mean cosine similarity between $w$ and each $b \in B$. The difference between these means is the relative association between each embedding $w \in X \cup Y$ and $A$ and $B$:

$$s(w, A, B) = mean_{a \in A} cos(w, a) - mean_{b \in B} cos(w, b)$$

For example, if $w$ were the embedding of a speech sample from a male speaker, this difference in means $s$ would measure how much more associated the sample was with positive valence than negative valence. These values for each of

the target stimuli are then aggregated to construct the effect size measurement, $d$. The SpEAT $d$ is given below, and shows how much closer the embeddings in $X$ are to $A$ than $B$, relative to the embeddings in $Y$. The effect size is the same as that used by Caliskan et al. (2017) and Steed and Caliskan (2021):

$$d = \frac{mean_{x \in X} s(x, A, B) - mean_{y \in Y} s(y, A, B)}{std\_dev_{w \in X \cup Y} s(w, A, B)}$$

Before cosine distances can be compared, however, embeddings of target and attribute stimuli need to be extracted. Pre-trained speech models, like some of the sentence encoders studied by May et al. (2019) with the Sentence Embedding Association Test (SEAT), create dynamically sized embeddings depending on the length of the sequence fed to the model as input (Radford et al., 2022; Hsu et al., 2021; Chen et al., 2022a; Baevski et al., 2020). Because embeddings are compared using cosine similarity in an EAT, which requires operands to have the same dimensions, this means that raw embeddings cannot be used directly. Another relevant feature of the pre-trained speech processing models we consider, also similar to many models tested with the SEAT, is that these models are multi-layered, with different layers providing information that is relevant to different downstream tasks (Radford et al., 2022; Hsu et al., 2021; Chen et al., 2022a; Baevski et al., 2020).

Recent approaches for adapting pre-trained models to specific tasks based on feature extraction rely on embeddings from all layers in the pre-trained model, rather than just a single layer (Yang et al., 2021; Peters et al., 2018, 2019). Intuitively, embeddings from internal layers may contain information not relevant to the task the model was pre-trained for, but important for the task the model is being adapted to. In speech, for example, embeddings from later layers in HuBERT and WavLM models have been shown to be more useful for granular tasks like ASR than embeddings from earlier layers, while for more speaker-centric tasks such as SID, embeddings from earlier in the models have been shown to be useful as well (Chen et al., 2022a). Because of the similarities in the models they study, as well as in the way that embeddings from the models are used, the SpEAT follows the SEAT in how it handles variable-sized, multi-layered embeddings (May et al., 2019). While the SEAT uses differing methods for aggregating embeddings for different

sentence encoders, however, we elect to apply the aggregation method they propose for ELMo (Peters et al., 2018) to all models we test. For each stimuli, this method involves taking the mean embedding within a layer, then summing these averaged embeddings together. We elect to use this aggregation method because the methods that May et al. (2019) propose for other models only use embeddings from a single layer in a model, which discards a large amount of information that is sometimes used when adapting pre-trained speech models to downstream tasks and can potentially contain bias (Yang et al., 2021; Chen et al., 2022a). We do test alternative strategies for extracting embeddings in Appendix D, however.

## 4 Data

We now describe the pre-trained models we focus on, their training data, and relevant features of the stimuli that we use for evaluating these models.

**Pre-Trained Models and Their Training Data** To study pre-trained models, we chose model families that are either widely used or achieve state-of-the-art performance on relevant speech benchmarks. We use multiple models from each family, rather than merely the largest model, as it is not clear that models from the same family encode the same biases, and multiple models from a family are often used by practitioners—not necessarily just the largest one. We use 16 models from these families: three from wav2vec 2.0, three from HuBERT, three from WavLM, and seven from Whisper. Further details concerning the models we use appear in Appendix B.

Of these models, wav2vec 2.0, HuBERT, and WavLM use Self-Supervised Learning (SSL) and train exclusively on speech, while models from the Whisper family, which use Weakly Supervised Learning (WSL), train on speech paired with transcripts. The wav2vec 2.0, HuBERT, and WavLM models we use are trained on either Librispeech (Panayotov et al., 2015) or Libri-Light (Kahn et al., 2020), speech datasets derived from readings of audio books recorded by volunteers as part of the LibriVox project. In addition to the Libri-Light training set, some WavLM models were also trained on additional data: 10,000 hours of audio from the GigaSpeech corpus (Chen et al., 2021), which is derived from podcasts, YouTube, and audiobooks, as well as 24,000 hours of audio from the VoxPopuli corpus (Wang et al., 2021a), which is derived

| Target Concepts X and Y | Corresponding IAT | Mean IAT $D$ | Data Source | Speech Content | N |
|---|---|---|---|---|---|
| Abled and Disabled | Visual IAT: (Nosek et al., 2007) | 0.45 | (Rudzicz et al., 2012) | Read Phrases | 60 |
| European-American and African-American | Visual IAT: (Nosek et al., 2007) | 0.37 | (Pitt et al., 2007), (Kendall and Farrington, 2021) | Extemporaneous Speech from Sociolinguistic Interviews | 60 |
| Female and Male | Audio IAT: (Mitchell et al., 2011) | 0.31 | (Weinberger, 2015) | Read Paragraph | 60 |
| Human and Synthesized | Audio IAT: (Mitchell et al., 2011) | 0.33 | (Weinberger, 2015), | Read or Synthesized Paragraph | 57 |
| U.S. and Foreign | Audio IAT: (Pantos and Perkins, 2012) | 0.32 | (Weinberger, 2015) | Read Paragraph | 59 |
| Young and Old | Visual IAT: (Nosek et al., 2007) | 0.49 | (Weinberger, 2015) | Read Paragraph | 58 |

Table 1: IATs used for evaluating whether the SpEAT captures human-like biases in pre-trained speech models. **X** and **Y** indicate the target concepts, **Data Source** identifies the source of the speech data used for representing target concepts, and **N** indicates the number of stimuli in each target group. **Mean IAT** $D$ indicates the mean IAT effect size measuring bias in humans, where a positive IAT $D$ value indicates that the X concept is more associated with positive valence than the Y concept in humans.

from European Parliament events. The Whisper models we use are either trained on a multilingual corpus, which consists of 680,000 hours of audio paired with transcripts in 97 distinct languages, or on the subset of the multilingual dataset which is in English.[3]

Weber (2021) performed an analysis of users who both had contributed to LibriVox and had catalog names that could be associated with gender, and identified 45.3% as female and 54.7% as male. Beyond this we are not aware of any information documenting the extent to which different speech styles are represented in the dataset, such as information on speaking styles. We note however that similar speech datasets have relied on U.S. speech. For example, of speech in the Common Voice English (7.0) dataset (Ardila et al., 2020) labeled by accent, about 47% was identified as U.S. English, compared to the next most frequent accent, England English, which made up about 16% of labeled speech (Markl, 2022).[4] The People's Speech dataset (Galvez et al., 2021), as another example, is identified by its authors as being made up primarily of American English. We hypothesize that U.S. English is also more prominent in the datasets that were used for training the models we consider, and following best practices for IATs that stimuli should be familiar to the entity being tested (Greenwald et al., 2022), we choose to center our work on U.S. English speech. We include results for U.K.-based tests in Appendix G however.

**Target and Attribute Stimuli** Following previous

EATs, we evaluate the SpEAT by testing for biases that have been found in humans using IATs and comparing the results in speech models to those found in humans (Table 1). We do so using two types of IATs: 1) foundational tests in the IAT literature, which were originally performed entirely using visual stimuli, and 2) audio IATs, which have explicitly established biases related to speech styles in humans (Mitchell et al., 2011; Pantos and Perkins, 2012). For visual IATs, we consider tests performed by Nosek et al. (2007) due to the large number of test participants (more than 20,000 people each). We focus on tests that use positive and negative valence as attribute concepts and that use target concepts for which there are speaking style differences between social groups, (for example we adapt an IAT related to age bias, but do not adapt the test related to weight bias). For audio IATs, we consider work from Mitchell et al. (2011) and Pantos and Perkins (2012). To our knowledge, these are the only peer-reviewed publications performing IATs that both use speech in Standard American English (SAE) and use positive and negative valence as attribute concepts.

Because Nosek et al. (2007) originally represented concepts visually, and because audio IATs from Mitchell et al. (2011) and Pantos and Perkins (2012) used short samples which may not fully capture variations in accents (for example recordings of speakers saying individual words), we adapt these tests by using alternate stimuli to represent the intended concepts.[5] To do so we follow guid-

---

[3]Transcripts in the multilingual dataset are either in the same language as the speech or in English.

[4]Common Voice has been used for training similar speech models such as XLSR (Conneau et al., 2020) and UniSpeech (Wang et al., 2021b).

[5]For audio IATs, in addition to SpEAT results for tests based on the adapted stimuli, we also include results based on the original stimuli that were tested on human subjects in Appendix E. We include an alternate set of stimuli for the Abled and Disabled test as well in Appendix E, which

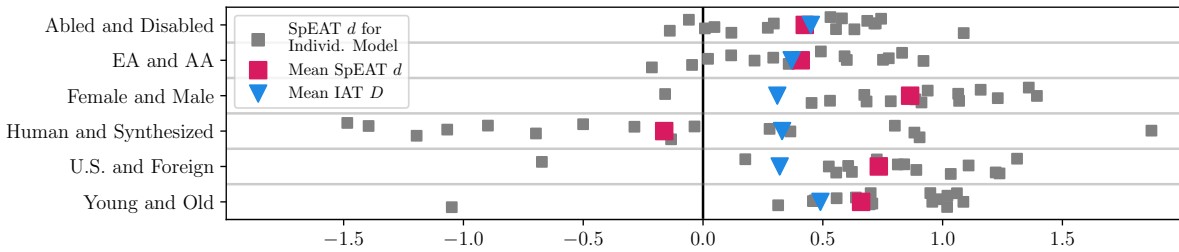

Figure 1: Each gray square corresponds to a bias test using embeddings from a single pre-trained model. Pink square refers to mean across all models, and blue triangle refers to mean IAT $D$ for bias measured in humans. A positive SpEAT $d$ indicates that in the model's embeddings, speech samples from the first target group (e.g. abled speakers or European-American (EA) speakers) are relatively more associated with positive valence over negative valence than samples from the second target group are (e.g. disabled speakers or African-American (AA) speakers). A SpEAT $d$ with the same sign as the IAT $D$ indicates that the speech model tends to favor the group that humans also tend to favor. 14 or more (out of 16) models show bias results that are congruent with biases found in humans regarding ability, race, gender, age, and accent.

ance from Greenwald et al. (2022) by selecting stimuli that are easily distinguishable. All stimuli that we use are in U.S. English (excluding the stimuli used to represent the concept of foreign accent). To ensure that differences between the stimuli representing the target concepts in a test are only due to differences in these concepts, and not due to another factor, we ensure that speech samples between the two target groups are matched. Stimuli are matched on gender as well as approximate age, with the exception of the female and male test, which is not matched on gender, the young and old test, which is not matched on age, the human and synthesized test, which is not matched on age,[6] and the abled and disabled test (for which age information was not made available for speakers). All tests, other than the test comparing speech from European-American speakers and from African-American speakers (which uses extemporaneous speech) are also matched in speech content, and involve phrases, sentences, or paragraphs that are repeated across conditions. Further details on the target stimuli chosen, as well as on the matching procedures used, can be found in Appendix C.[7]

For attribute stimuli, we use speech from the Morgan Emotional Speech Set (MESS) (Morgan, 2019), a corpus of 1,800 semantically neutral recordings made by six Caucasian actors, aged 19-21. The Morgan Emotional Speech Set is the only publicly-available corpus of which we are aware that contains SAE speech rated on valence. Voice actors were native speakers of American English, evenly split on gender, and were asked to read sentences while portraying one of four categorical emotions—happy, sad, angry, or calm. The recordings were then rated on valence by 10 listeners who were aged 19-28 and evenly split on gender. To ensure that differences in association with positive and negative valence are not due to differences in the speakers whose speech is used to represent the valenced concepts, we ensure that each speaker is represented equally in each valence pole. We select the 10 recordings from each speaker that have the highest average valence ratings to represent positively valenced speech, and the 10 recordings from each speaker that have the lowest average valence ratings to represent negatively valenced

---

represents the target concepts using short phrases.

[6]Synthesized speech is generated with Microsoft SAPI 4.0, which contains the synthetic voices originally employed for evaluating bias in humans. We use this generator in order to ensure that the biases we measure in speech models relate to the same concepts as the biases we compare to in humans. Each voice is Microsoft SAPI 4.0 is labeled with a gender or a gendered name that the voice is intended to represent, and we use these labels to ensure that the set of speech samples we use is balanced on gender. As of submission, code for the generator is available at https://github.com/TETYYS/SAPI4.

[7]To ensure that differences in associations with valence in the European-American and African-American test are not due to differences in semantics, we test whether the speech content from one social group tends to show more positive valence

than speech content from the other. We pass transcripts for the speech from European-American and African-American speakers through five commonly used sentiment analysis tools (Barbieri et al., 2020, 2022; Loureiro et al., 2022; Hutto and Gilbert, 2014; Elias, 2022), then use a Welch's unequal variances $t$-test to evaluate whether the sentiment scores differ by race on average (Delacre et al., 2017; West, 2021). Each sentiment analysis tool gives three scores (negative sentiment, neutral sentiment, and positive sentiment); VADER also gives a compound score. Using the typical $p = 0.05$ threshold, across all 16 scores output by these models, only one significant difference is found: the positive sentiment score given by the model from Loureiro et al. (2022) ($p = 0.02$). In this case, however, the social group that is scored higher on positive sentiment for semantics is not the one that tends to be favored by the speech processing models.

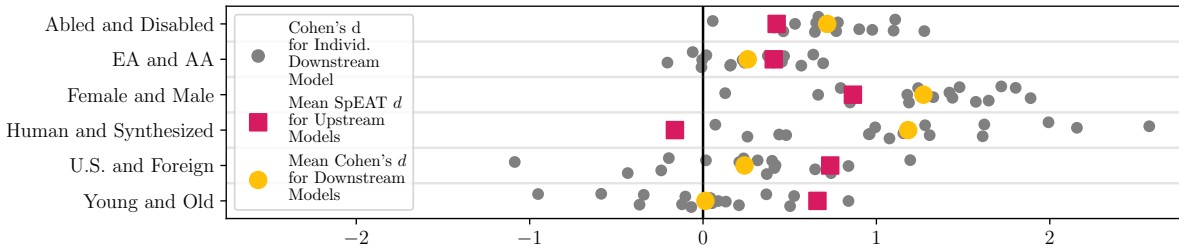

Figure 2: Comparison of bias found in pre-trained model using SpEAT to bias found in downstream SER model. Gray circles correspond to a bias test using an individual downstream SER model, and yellow circles show mean bias across downstream SER models. In 66 of the 96 (69%) tests performed, the group more associated with positive valence by the pre-trained model is also more associated with positive valence by the downstream SER model.

speech. This results in 60 samples representing positive valence and 60 samples representing negative valence.

## 5 Evaluation and Results

We provide three evaluations of the SpEAT. **First,** we test whether it captures human-like biases in speech models by comparing results found in pre-trained models using the SpEAT to results found in humans using IATs. **Second,** as the SpEAT is designed for pre-trained models that are often adapted to other tasks, we also evaluate the method by testing whether these biases propagate, and whether biases found in models using the SpEAT are indicative of biases in downstream models. **Third,** we measure how much uncertainty is associated with the SpEAT effect size by studying how its SE varies when different numbers of stimuli are used to represent target concepts.

**Comparison to Human Biases** Similar to other EATs, our first evaluation of the SpEAT is to test whether it detects biases in speech models using ground truth data from human biases (Caliskan et al., 2017; Steed and Caliskan, 2021). We compare the direction of the SpEAT $d$ to the direction of the mean IAT $D$ among humans for a number of well documented biases to check for stereotype-congruence. For each speech model discussed in Section 4, we calculate the SpEAT $d$ for each bias in Table 1. All IAT $D$ values we compare to are positive, meaning that humans showed more of an association between the first target group and positive valence than between the second target group and positive valence. SpEAT $d$ values for the 16 models we test are displayed in Figure 1. If the SpEAT were able to find human-like biases in pre-trained models in speech, we would expect SpEAT $d$ values to be identical in sign to the mean

IAT $D$ values given in Table 1. All IAT $D$ values that we compare to are positive, and range between 0.31 and 0.49. Of the tests we perform, 79 out of 96 (82%) have positive SpEAT $d$ values, reflecting the same biases recorded in society. Results vary depending on the test, however: only 6 of 16 models show bias against synthetic voices, while 14 or more models show stereotype-congruent biases for all other tests.[8]

There does appear to be a slight model family effect, with models in the same family having somewhat of a tendency toward similar SpEAT $d$ values concerning a given social group. Performing a two-way ANOVA with the SpEAT $d$ value as dependent variable and social group and model family as independent variables, we find that the interaction between model family and social group is just below the typical p=0.05 significance threshold, (p=0.048). The main effect of model family is not significant, however, suggesting that it is not the case that SpEAT $d$ values for a family tend to be located in the same area across all social groups. For example, Whisper SpEAT $d$ values tend to be similar to one another in the African-American and European-American test, but Whisper SpEAT $d$ values for the African-American and European-American test do not tend to be similar to Whisper SpEAT $d$ values from the Young and Old test.

**Bias Propagation to Speech Emotion Recognition** We further evaluate the SpEAT by testing whether SpEAT $d$ values for a pre-trained model are related to bias in the downstream task of SER, which involves predicting emotions associated with speech (Mohammad, 2022). Using an architecture from Yang et al. (2021) designed for adapting pre-

---

[8]Previous EATs have also reported $p$-values for a related Null Hypothesis Significance Test, which we also report in Appendix A.

trained models to downstream tasks, we adapt each of the 16 pre-trained models to predicting valence in the full MESS dataset (consisting of 1,800 samples) which is the only publicly available corpus of SAE labeled for valence of which we are aware. We train three separate downstream models for each pre-trained model.[9] We then predict the valence of speech stimuli that were introduced in Table 1, for example stimuli from abled and disabled speakers, and calculate Cohen's $d$ for the predicted valences for each pair of target concepts. We consider differences in standardized mean predicted valence between the two groups to be a bias due to the matched nature of the stimuli.

As shown in Figure 2, we find that SpEAT $d$ values favoring one target group tend to align with speech from the favored target group being predicted as higher in valence: In 66 of the 96 (69%) tests performed, the group more associated with positive valence by the pre-trained model is also more associated with positive valence by the downstream SER model. We also note that in all tests excluding that related to human and synthesized speech, the average Cohen's $d$ across all 16 speech models is in the same direction as the average SpEAT $d$ across the models.

**Uncertainty of SpEAT $d$** Our final evaluation of the SpEAT involves quantifying the uncertainty associated with SpEAT $d$ values when different numbers of stimuli are used to represent target concepts, in order to evaluate the extent to which SpEAT effect sizes might differ if they were recalculated based on similar datasets. We do so by estimating the SE for each SpEAT $d$ value at varying sample sizes. The SE of a statistic measures how much the statistic will vary if it is calculated repeatedly based on new data. Lower values indicate that the statistic will vary less, and that there is less associated uncertainty. We estimate the SE using bootstrapping (Hesterberg, 2011), where we sample with replacement from the original dataset, repeatedly calculate the statistic, then calculate the sample standard deviation of these bootstrapped statistics. We perform resampling to match the original sampling strategy used for each dataset, (for example, in the abled and disabled test where each individual stimuli from a disabled speaker was matched with an individual stimuli from an abled speaker saying the same phrase, we resample by pair) (Hesterberg,

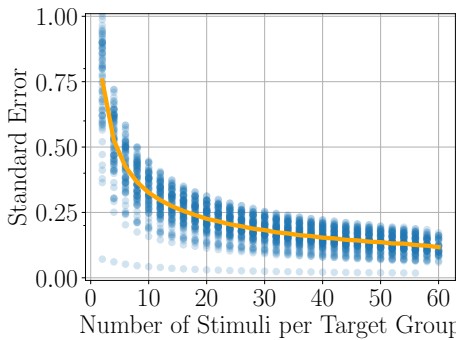

Figure 3: SE for SpEAT $d$ when different numbers of stimuli are used to represent each target concept. Lower SE indicates a more precise estimate. Each point represents a single bootstrap estimate based on a single number of target stimuli, type of bias, and speech model. Orange line indicates average SE at each number of stimuli (across types of bias and speech models).

2011). For each bias test and speech model, we calculate multiple bootstrap estimates of the SE, using different numbers of target stimuli for each. This allows us to study how the SpEAT's uncertainty changes when more or less stimuli are used to represent the target concepts. Each SE estimate is based on 10,000 bootstrap resamplings. Results in Figure 3 show that the SE decreases sharply as the sample size increases: the average standard error when two stimuli are used per target group is 0.75, the average standard error when 10 stimuli are used is 0.33, and the average standard error when 60 stimuli are used is 0.12.

## 6 Discussion

The results in Figure 1 show that pre-trained speech processing models trained on large corpora of speech data, like those trained on large corpora of image or text data, can easily learn human-like biases related to gender, race, ability, age, and accent. Work in computer vision has suggested that biases detected by EATs may arise from co-occurrence in the training data (Steed and Caliskan, 2021), and we hypothesize that co-occurrence may contribute here as well, for example if speakers who use one speech style tend to more frequently express positive valence than speakers who use other speech styles in the training set. We also hypothesize that the wide dispersion in the test comparing human and synthesized speech may be related to frequency as it is not clear that synthesized speech is contained in any of the pre-training corpora that the models we consider were trained on.

---

[9]Full details on the training process for downstream models is provided in Appendix F.

Beyond just identifying that biases can be learned from data in a new modality, however, this work also identifies a new means of representing target concepts. In using speaking styles to represent concepts, rather than names or words as in text-based models, we've shown that not only do models demonstrate bias based on language content, but also based on how, and by whom, it is delivered. The data used for training models in computer vision, natural language processing, and speech processing is inherently multi-faceted—it does not connect to social groups in just one way. This data is produced by people who belong to certain social groups; it may be about people of certain social groups; it may be recorded by people of certain social groups; and it may also have meaning to people of certain social groups. Our results suggest that, at least in speech processing, models are able to learn associations beyond just content, and raises the question of whether there are differential associations related to the provenance of text and image data as well. Would text of similar meaning, but written by people of different social groups, have different associations with valence?

Beyond adding further evidence for bias in AER (Kiritchenko and Mohammad, 2018), our experiment with SER models also adds evidence that biases in pre-trained models can propagate downstream. The biases that we test for in upstream and downstream models are connected in that they both involve valence—in the upstream model the SpEAT tests for associations with valence, while in the downstream model we measure bias using standardized differences in predicted valence between speech samples from people of differing social groups (which have been matched using the procedure in Section 4). Studying the relationship between intrinsic bias in embeddings measured using an EAT and performance bias in downstream tasks using a variety of performance metrics, tasks, and models, Goldfarb-Tarrant et al. (2021) find that the relationship between the two biases is mostly uncorrelated. They do, however, find that one of two models they test shows a slightly positive relationship between intrinsic bias and bias in a conceptually related downstream task. Work by Steed et al. (2022) similarly finds a slight connection between upstream and downstream bias in one of the two tasks they test, and Ladhak et al. (2023) find that when considering name-nationality bias, abstractive upstream models can pass bias on to a down-

stream task. Considering these works together, our results add more evidence that there can be a relationship between upstream and downstream bias when the two biases are related in concept.

Our work calls attention to the need for detailed study of bias mitigation techniques in pre-trained speech processing models. While we are not aware of any technical solutions for mitigating *implicit* bias in pre-trained speech models, work by Meng et al. (2022) has tested the effect of the makeup of the training set on performance biases observed in downstream tasks. Their results suggest that altering dataset composition is likely not a complete solution to bias in pre-trained models, however. It is possible that methods that have previously been proposed for mitigating bias in other modalities may be useful for managing the bias that has been shown in speech. Strategies that have been explored in computer vision and natural language processing include oversampling the training set to make it more representative of social groups; adversarial training, whereby in learning a task the model also learns to explicitly ignore demographic information; and embedding modification, whereby the geometry of the embeddings is altered in an attempt to remove information related to demographics (Wang et al., 2020; Sun et al., 2019).

# 7   Conclusion

In this paper we contributed the SpEAT, a novel method for measuring bias in pre-trained speech processing models. We used the SpEAT to study state-of-the-art pre-trained speech models which are likely to have a wide impact on a variety of speech processing tasks. We tested for six categories of bias in 16 pre-trained models, and found consistent biases in how speech samples from speakers of different ages, races, genders, and abilities were associated with speech samples rated on valence by human listeners. We also showed that biases detected with the SpEAT propagated to the downstream task of SER, as a pre-trained model displaying bias toward a group was correlated with bias in downstream models adapted to SER. These findings imply, first, that association-based biases are present in a larger number of modalities than had previously been known, and second, that upstream bias may propagate to downstream bias, when the two biases are connected.

## 8 Limitations

In designing our work, we elected to focus on state-of-the-art and popular speech processing models, and to validate the SpEAT using biases relating to speech styles that have previously been documented in humans with IATs. The biases we studied concern social characteristics with wide applicability, such as race, gender, age, and ability, however they do not capture the full extent of biases that may be contained in speech models. While the SpEAT does allow for the future study of biases related to a wider set of social groups, more consideration and data will be needed for adapting the test to quantify biases related to, for example, a wider range of gender identities or ethnicities.

Our results may be impacted by the 120 speech samples that were used to represent positive and negative valence, which, while balanced on gender, was provided entirely by six younger Caucasian speakers and rated entirely by 10 younger listeners whose race information was not given (Morgan, 2019). As positive and negative valence were *both* represented by speech from the same speakers (and rated by the same listeners), it is at least clear that race and age-based associations are not due to differences in the races and ages of speakers used for each valence pole. While we are not aware of a publicly-available dataset of SAE with the necessary information to evaluate whether the SpEAT $d$ is affected by the demographics of people used to create the attribute stimuli, we look forward to future work in this direction.

## 9 Ethics Statement

The central focus of this work is evaluating the extent to which pre-trained speech processing models contain intrinsic biases and if these biases may propagate downstream. We recognize that metrics are not solutions in and of themselves, and can even be used to propagate bias if deployed in a harmful manner (for example if practitioners measure bias and then purposefully select biased models for downstream applications). We believe, however, that detecting and measuring bias plays an important role in raising awareness, avoiding harmful impacts, and studying the science of bias.

While we study the connection between bias in a pre-trained model and bias after the model has been adapted to SER, we would like to note that bias is just one of many potential issues with SER. SER may purport to capture the inner state of the speaker, but is at best only able to capture the emotion being conveyed, or the emotion being perceived by the annotator (Mohammad, 2022). Furthermore, even well defined and accurate SER runs the risk of setting and enforcing norms for how emotions should be expressed, being used for emotional manipulation, and degrading privacy and autonomy.

## Acknowledgments

We would like to thank Hari Iyer, along with the anonymous reviewers, for their comments and suggestions. This work was supported by the U.S. National Institute of Standards and Technology (NIST) Grant 60NANB23D194. Any opinions, findings, and conclusions or recommendations expressed in this material are those of the authors and do not necessarily reflect those of NIST. This work was facilitated in part through the use of advanced computational, storage, and networking infrastructure provided by the Hyak supercomputer system and funded by the STF at the University of Washington.

## Disclaimer

These results presented in this paper are not to be construed or represented as endorsements of any participant's system, methods, or commercial product, or as official findings on the part of NIST or the U.S. Government.

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

# A    Statistical Significance of SpEATs

Along with an effect size metric $d$, many prior works concerning EATs also include results from a Null Hypothesis Significance Test (NHST) (Caliskan et al., 2017; Steed and Caliskan, 2021; May et al., 2019). This NHST tests the null hypothesis that the EAT $d$ for a model and set of concepts is equal to 0 (or equivalently that stimuli representing the first target concept and stimuli representing the second target concept are equal in terms of their relative distances to stimuli representing the two attribute concepts). The NHST tests this null hypothesis against the alternate hypothesis that the EAT $d$ is greater than 0 for a model and set of concepts.

We include results from this NHST in Figure 4. We find that 63 of 96 tests (66%) are significant at the $\alpha = 0.01$ level. Because performing such a large number of NHSTs in unison increases the probability of accidentally rejecting a true null hypothesis at least once, we also perform a Bonferroni correction for multiple comparisons. Bonferroni corrections make the threshold required to claim significance in individual tests more strict, lowering the probability of accidentally rejecting a true null hypothesis in any individual test. We find that 46 of the 63 tests (73%) that were significant at the $\alpha = 0.01$ level remain significant after the correction.

# B    Further Details of Pre-Trained Speech Models

Below, we provide further details concerning the pre-trained speech models that we evaluate bias in.

## B.1    wav2vec 2.0

We test wav2vec 2.0 (Baevski et al., 2020), a speech representation learning framework originally published in 2020, due to its wide usage. As of submission in June of 2023, wav2vec 2.0-based models make up seven of the ten most downloaded ASR models over the last month in HuggingFace's transformers library. In the wav2vec 2.0 family, we use the non-finetuned wav2vec 2.0 Base and wav2vec 2.0 Large (both trained on Librispeech (Panayotov et al., 2015)), as well as the non-finetuned wav2vec 2.0 Large (LV-60), (which was trained on Libri-Light (Kahn et al., 2020)). During the pre-training phase, wav2vec 2.0 models are trained using a contrastive loss to identify continuous sequences of audio. The model takes wav files as input, splits

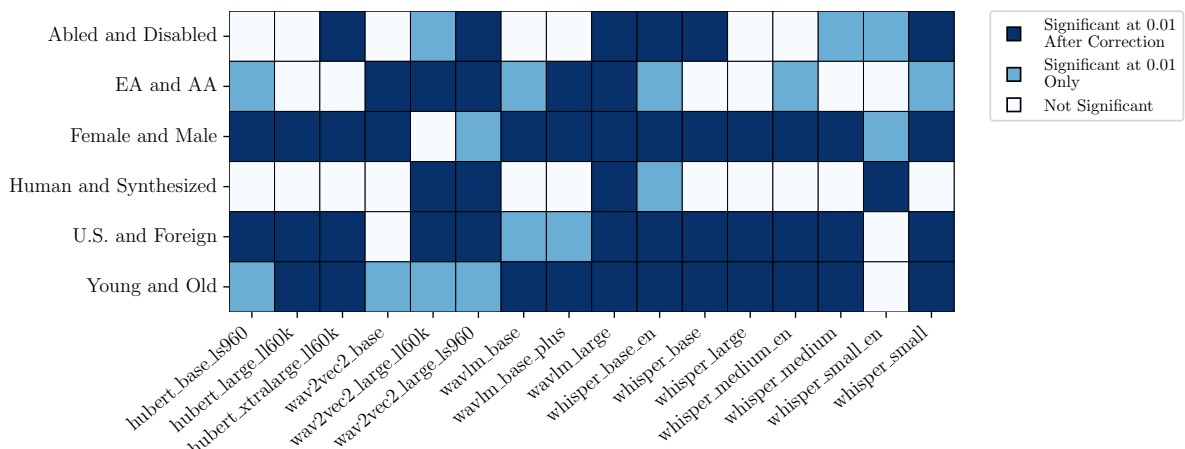

Figure 4: Statistical significance of SpEATs. We find that 63 of 96 tests (66%) are significant at the $\alpha = 0.01$ level. Of the tests significant at the $\alpha = 0.01$ level, 46 (73%) are also significant after a Bonferroni correction for multiple comparisons.

them into a sequence of chunks, embeds each of the chunks in latent space using a set of feature encoding layers, then masks some elements in the sequence of latent representations. The latent representations are contextualized using a set of sequence encoding layers, and the model is then tasked with identifying the correct masked chunk of sound from a set of random distractor samples, using the chunks that came prior in the sequence as context.

Architecturally, the feature encoding portion of the model, which operates on each element in the sequence of audio chunks individually, contains seven blocks. Each block contains a temporal convolution step, a layer normalization step, and a Gaussian Error Linear Unit (GELU) activation, which use 512 channels, strides of (5,2,2,2,2,2,2) and kernel widths of (10,3,3,3,3,2,2). The sequence encoder portion contains transformer blocks - 24 blocks in the largest version of the model, each with 16 attention heads. We take embeddings from the end of each transformer block (as well as from before the first transformer block).

## B.2 HuBERT

Hidden-Unit BERT (Hsu et al., 2021) (HuBERT) is a current open source state-of-the-art framework for learning representations of speech data. A fine-tuned HuBERT model achieves the first-best and third-best score of all open source models on two widely-used benchmarks for measuring performance in ASR, according to Papers With Code. These benchmarks, the LibriSpeech (Panayotov et al., 2015) test-other and test-clean datasets, measure the Word Error Rate (WER) for a clean speech set and a noisy speech set, respectively. (WER being the mean edit distance, at the word level, between predicted and true transcriptions). In the HuBERT family we use all available pre-trained models that were not finetuned: Base, Large, and Extra Large. The Base version of HuBERT was trained using Librispeech, while the larger versions were trained using Libri-Light. In addition to its performance on LibriSpeech, HuBERT Large is currently ranked only beneath WavLM models in the public SUPERB Challenge, which measures speech representation models on how well they adapt to a battery of downstream tasks, including ASR, ST, and speaker diarization.[10]

HuBERT inherits much of its architecture from prior unsupervised speech models, but also incorporates a pseudo-labeling component adopted from the vision model DeepCluster (Caron et al., 2018). Like its predecessors wav2vec (Schneider et al., 2019) and wav2vec 2.0 (Baevski et al., 2020), HuBERT contains a feature encoder portion, for learning individual representations of sequenced snippets of sound, followed by a sequence encoder portion, used to build contextualized representations from the individual representations. Rather than try to predict masked representations directly, HuBERT's loss involves the prediction of psuedo-labels, which it creates itself, using $k$-means clustering. Unlike wav2vec and wav2vec 2.0 models,

---

HuBERT uses cross-entropy loss for pre-training, to predict the psuedo-label that the next snippet belongs to. It alternates between generating clusters and predicting those clusters.

Each layer in the feature encoder blocks of HuBERT consists of a temporal convolution, a layer normalization (Ba et al., 2016), and a GELU activation (Hendrycks and Gimpel, 2016). There are 7 layers in the feature encoder portion, each of which has 512 channels. The sequence encoder portion of HuBERT models contain between 12 and 48 transformer blocks and between 8 and 16 attention heads, depending on the model size (Hsu et al., 2021). During pre-training, a projection layer is applied to the output from the sequence encoder portion of the model, as well as a code embedding layer. For extracting representations from HuBERT models, we take embeddings from the end of each transformer block, relying on the embedding extraction implementation published with the original paper.

### B.3 WavLM

WavLM (Chen et al., 2022a) is another family of state-of-the-art representation learning models for speech data. As of this paper's submission WavLM is ranked first on the public SUPERB Challenge, implying that its pre-trained weights adapt well to a wide variety of speech tasks. In the WavLM family, we also use all available versions: Base, Base+, and Large. WavLM Base uses Libri-Light for training, however the WavLM Base+ and Large versions extend the training data to GigaSpeech and VoxPopuli. WavLM is based on HuBERT, but uses a different training setup and slightly different architecture. In addition to the extended dataset, WavLM models also employ a data pre-processing technique whereby 20 percent of pre-training inputs are mixed with other noise, while their labels from the $k$-means portion of the model are left untouched, to make the model more robust. Architecturally, WavLM models involve gated relative position biases that are incorporated into their transformer blocks, whereas HuBERT models use convolutional relative position embeddings. We take embeddings from the end of each transformer block (as well as from before the first transformer block). We use the embedding extraction implementation published with the original paper.

### B.4 Whisper

The final model family we test is Whisper (Radford et al., 2022). While Whisper does not outperform wav2vec 2.0 on the LibriSpeech test-clean benchmark, it does outperform wav2vec 2.0 at transcribing a wide array of other datasets, implying Whisper may be more robust. Whisper models can be used directly for ST, spoken language identification, voice activity detection, and multilingual ASR. In the Whisper model family, we use the Base, Small, Medium, and Large versions, as well as any English-only variants used for these models. We use the 1.0 versions of these models, which were the only versions available when our experiments were run.

A difference between Whisper and the other model families tested is that Whisper uses a sequence-to-sequence encoder-decoder transformer architecture. Log-mel spectrograms are first created from audio inputs, which are then fed into convolutional layers and a GELU activation layer (Hendrycks and Gimpel, 2016), before being passed on to between four and 32 transformer encoder blocks (Vaswani et al., 2017), depending on the model size. The decoder portion of the model contains between four and 32 transformer decoder blocks as well. We take embeddings from after each of the transformer encoder blocks. We test both the multilingual and English-only variants of the Base, Small, and Medium versions, as well as the multilingual Large version (which does not have an English-Only variant).[11]

## C Further Details on Target Stimuli

We now provide details on the stimuli used for representing target concepts.

### C.1 Abled and Disabled Speakers

We adapt the test performed by Nosek et al. (2007), originally using images representing abled and disabled people, to speech. We elect to represent disability and ability using speech from dysarthric and non-dysarthric speakers, respectively. Dysarthria is a speech condition caused by muscle weakness, that is considered to be noticeably low in valence to human listeners (Lass et al., 1988). While dysarthric speakers do not represent all disabled people, IAT stimuli should be easily distinguishable across target groups (Greenwald et al., 2022),

---

[11]For clarity, despite using the multilingual versions of the models, we only use English stimuli for our tests.

and we choose dysarthria as one relevant speech condition, for which speech data is readily available. We leave testing of other forms of disability that manifest in speech to future work.

We use speech from the TORGO Database to represent speech from disabled and abled speakers. The database contains speech samples from eight dysarthric and seven non-dysarthric speakers, who are approximately equal in mean age across conditions. Speakers were given a variety of prompts for different stimuli, including single words, sentences designed to elicit accent differences, as well as images used to elicit natural and unrestricted sentences. In order to capture speech style differences, we use speech samples that contain two or more words. We match samples in gender and speech content across conditions, but are unable to match in age, as ages of individual speakers were not included with the dataset.

To test the extent to which results from the TORGO Database extend to other datasets, we also carry out the SpEAT using speech from the Universal Access Speech (UASpeech) dataset (Kim et al., 2008) and present the results in Appendix E. We use the most recently cleaned version of UASpeech, which was processed in 2020 using noisereduce. UASpeech consists of recordings of words read by dysarthric and non-dysarthric speakers. The UASpeech set is the largest dataset of which we are aware that contains matched English speech from dysarthric and non-dysarthric speakers, however because recordings in the UASpeech are of single words or phrases, rather than complete sentences, which may not fully capture accent differences across conditions, we elect to only present results from the TORGO database in the main body of the paper. Speakers were paired across conditions based on approximate age in UASpeech (being within three years in age of each other), as well as by gender. As the corpus contains more than 100,000 recordings in total, we elect to sample the number of recordings to be on par with the sample sizes used for EATs performed in other modalities. We randomly sample five words per speaker pair, which gives 55 samples to represent each group.

## C.2   European-American Speakers and African-American Speakers

We also adapt the test by Nosek et al. (2007) of the concepts of European-American and African-American to the speech modality. We take speech data from two separate datasets to do so: The first is the Buckeye Corpus (Pitt et al., 2007), which contains speech entirely from Caucasian speakers, and the second is the Corpus of Regional African American Language (CORAAL) (Kendall and Farrington, 2021), which contains speech entirely from speakers of African-American Language. Both datasets consist of sociolinguistic interviews of variable length, recorded on high quality equipment.

We start with the full Buckeye Corpus, which was published in 2007, as well as with all CORAAL components that were recorded after the year 2000 and available in October of 2022. We process the datasets by isolating speech from the interview subjects, removing any non-speech noises, and removing any clips from speakers who are younger than 18 years of age. We match by speaker gender, interviewer gender, and speaker age. (The Buckeye Corpus does not contain an exact age, only information on whether the speaker is younger than 30 or older than 40, so we calculate the same feature in CORAAL and use this as our age feature.)

Like UASpeech, CORAAL and Buckeye both contain many more recordings than have been used in EATs in other modalities, so to conserve resources we again sample the datasets. We match by gender and approximate age, then take a balanced stratified sample, (by speaker age and speaker gender), to ensure that no age or gender groups are over or under represented in our sample, and the SpEAT scores do not pertain to speakers of any specifc age or gender group more than another. We take 15 samples per cell, which gives us 60 samples from speakers of each ethnicity. We perform a paired samples $t$-test to ensure that audio clips do not differ in length on average, in order to establish that any potential differences in association are not simply due to differences in audio clip length. A $t$-test is appropriate due to the sample size of the dataset, and a paired samples test is required due to the matched relationship between the audio clips. We do not find evidence that clips between the two groups differ in length on average ($t(59) = -0.12$, $p = 0.90$), which implies that differences in association are not attributable to differences in length of the input audio clips.

## C.3   Female and Male Speakers

We also adapt audio IATs from (Mitchell et al., 2011) for comparing female and male speech. We

use speech from the Speech Accent Archive (SAA), a corpus containing over 2,800 samples of English speech from unique speakers, as well as metadata on the speakers' ages (Weinberger, 2015). Data in the archive consists of high quality recordings of speakers reading an identical elicitation paragraph, and range between 10 and 80 seconds long. While the SAA does not contain a label indicating whether speech is in SAE, it does contain information on speakers' places of birth, and in order to ensure that the speech we use is representative of SAE, (which was used for the original tests performed on humans), we limit our sample to speakers whose place of birth is in the United States. (Speakers must also have English listed as their native language.) We match by age across conditions.

To test the extent to which results from the SAA generalize to other datasets, we also perform the female and male test using the speech samples that were originally shown to humans in the audio IATs performed by Mitchell et al. (2011). We present these results in Appendix E. The data for these tests come from four distinct human speakers and four distinct synthesized voices: two human female speakers, two human male speakers, two synthetic female voices, and two synthetic male voices. The synthetic voices come from Microsoft and Read-Please. Each speaker or synthetic voice is used to generate speech from the same 16 short neutral phrases, such as "candle holder" or "cardboard box." We test 64 samples per target group.

### C.4 Human Speakers and Synthesized Speech

We also adapt the test from Mitchell et al. (2011) comparing human and synthesized speech using speech from the SAA. Text to Speech technology has changed significantly since Mitchell et al. (2011) carried out IATs using synthetic voices, and to ensure that the synthetic voices that we use are similar to the synthetic voices that were originally played for humans, we use similar Text to Speech systems to those used originally. The original voices used were Microsoft Mary, Microsoft Mike, as well as the ReadPlease Male and Female voices. These voices are included in Microsoft Speech API 4.0 (SAPI4.0), from which we also include 15 other synthetic voices in our tests. We generate speech using the same elicitation text as that used for human speakers in the SAA, using an

online generator.[12] In addition to samples created using the default speeds provided by the generator, we also create fast and slow versions of each voice, where the fast version has a speed parameter 1.25 times the default value, and the slow version has a speed parameter 0.75 times the default. These synthetic samples do not have associated ages, but do have associated genders, which we are able to use for matching the synthetically generated samples to speech from the SAA. (Because human speakers in the original IAT were speakers of U.S. English, we filter to only include speakers whose native language is English and who were born in the United States.)

To test whether results from the SAA generalize to other datasets, we also perform the human speaker and synthesized speech test using the same stimuli as (Mitchell et al., 2011). We present these results in Appendix E. Stimuli are the same samples consisting of single words or phrases as those used in Appendix C.3, however now split by whether the sample contains speech from a human speaker or from a synthetic voice, rather than by gender. We test 64 samples per target group.

### C.5 United States and Foreign Accented Speakers

We also adapt the audio IATs performed by Pantos and Perkins (2012), which compare English language speech from a native speaker of United States English to speech from a native speaker of Korean. Korean was originally chosen to represent the concept of foreign accent in order to minimize the associations that listeners might have with stereotypes, as prior work had found that Korean accents were difficult to identify in the United States (Lindemann, 2003). We use speech from the SAA to represent both U.S. and Foreign speakers, and use similar filtering criteria to those used by Pantos and Perkins (2012) to select speakers. We use speakers whose native language is English and whose place of birth is in the United States to represent U.S. accented speech, and speakers whose native language is Korean and whose place of birth is in Korea to represent Korean accented speech. Samples were matched on age and gender across conditions.

To evaluate whether results from the SAA extend to other datasets, we validate the test results

---

[12]As of submission, code for the generator is available at https://github.com/TETYYS/SAPI4.

| | | Layer Agg. | | | | | | | | | |
|---|---|---|---|---|---|---|---|---|---|---|---|
| | | Sum | Min | Max | *First* | *Second* | *Q1* | *Q2* | *Q3* | *Penultimate* | *Last* |
| Temporal Agg. | Mean | 0.82 | 0.72 | 0.70 | 0.52 | 0.69 | 0.80 | 0.92 | 0.88 | 0.76 | 0.73 |
| | Min | 0.66 | 0.66 | 0.71 | 0.61 | 0.80 | 0.77 | 0.74 | 0.65 | 0.64 | 0.57 |
| | Max | 0.66 | 0.56 | 0.68 | 0.47 | 0.70 | 0.82 | 0.69 | 0.65 | 0.68 | 0.59 |

Table 2: Proportion of SpEAT $d$s that are positive, and therefore congruent with biases that tend to be found in humans, when different strategies are used for aggregating raw embeddings. Temporal Agg. refers to the first step of aggregation—summarizing across the variably-sized sequence dimension of the raw embeddings (whose length depends on the duration of the input speech sample). Layer Agg. refers to the second step—aggregating across Transformer layers. For aggregation across layers, aggregation strategies other than sum, min, and max involve taking only embeddings from an individual layer, for example Q2 refers to selecting embeddings from the median layer in the model (e.g., layer 24 in the 48 layer HuBERT XL). Layer aggregation strategies that involve selecting from a single layer are denoted in italics.

using the original stimuli played to humans. The original tests use speech samples read by two similarly aged male actors in their native accents as target stimuli. The actors read the same eight neutral phrases, for example "at this point", and the phrases are repeated three times for human listeners. We use the exact samples shown to humans, (i.e. those that repeat the phrases three times).

### C.6 Young and Old Speakers

We also adapt the IATs performed in (Nosek et al., 2007) comparing the concepts of young and old. We use speech from the SAA, and limit ourselves to speakers whose birth place is in the United States. We take the oldest and youngest speakers to represent the concepts of young and old. To ensure that our results are not more applicable to male or to female speakers, we balance the samples based on gender. We take 58 speakers per target group (29 young male speakers, 29 young female speakers, and so on). The young category consists of speakers between ages 18 and 19, while the old category consists of speakers between ages 57 and 93.

### D Other Embedding Aggregation Strategies

Similar to work from May et al. (2019), who test various sequence and layer aggregation in text-based models, we evaluate whether our results would have changed had a different strategy been used for summarizing embeddings, (other than using the mean across the temporal dimension then the sum across layers). The proportion of SpEAT $d$s that are positive are shown in Table 2. All but one of the aggregation strategies result in more positive SpEAT $d$ values than negative, however

there is some variability between strategies. Taking the mean across the temporal dimension tends to result in more alignment with human biases than taking the min or max, for example, and selecting embeddings from early to middle layers also tends to result in more alignment with human biases.

### E SpEAT Results Using Alternate Stimuli Sets

As described in Appendix C, we also perform EATs using alternate stimuli, to test the extent to which the results we find would generalize to other datasets. The SpEATs comparing U.S. and Foreign speech using the stimuli originally used by Pantos and Perkins (2012) are opposite in mean from the tests comparing U.S. and foreign speech using speech from the SAA, however other tests are similar in mean when comparing original to alternate stimuli sets. We hypothesize that this may be related to the number of stimuli used by Pantos and Perkins (2012) to represent each target concept—8 stimuli, rather than the more than 50 stimuli that were used for all other tests performed—or related to the fact that speech samples are from two speakers (one to represent U.S. speech and one to represent Foreign speech), and contain single words or phrases repeated three times, rather than lengthy and fluid samples such as that contained in the SAA.

### F Downstream Speech Emotion Recognition Models

Here we describe the training process for the models used for the downstream task of SER, which is taken from Yang et al. (2021). After extraction

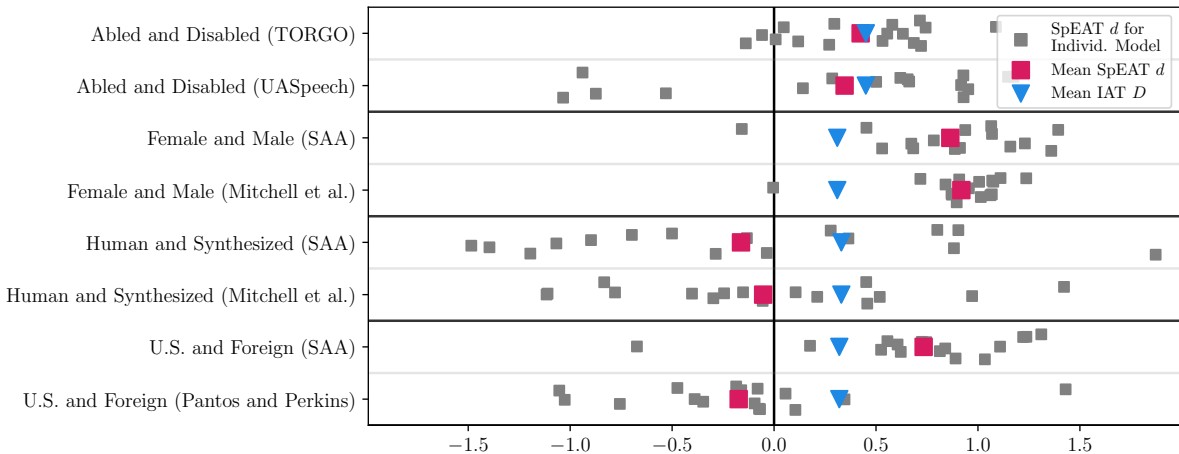

Figure 5: SpEAT $d$ values showing the extent to which results extend to other datasets that can represent the same concepts. Speech samples in original datasets (TORGO and SAA) are longer and include more words, and were hypothesized to better capture differences in speaking style. Three of four tests using alternate stimuli show similar mean SpEAT values to original tests, while the fourth (which uses a small number of samples, only includes speech from a single speaker to represent each target concepts, and contains speech samples consisting of single words or phrases repeated three times) does not.

from the upstream pre-trained model, each embedding starts as shape (N Layers, Seq Len, Embd Dim). The first dimension (N Layers) corresponds to the number of layers in the pre-trained model. The second dimension (Seq Len) corresponds to the length of the input speech sample. The third dimension (Embd Dim) corresponds to the width of the pre-trained model. The downstream model first takes a weighted average over the first dimension of the embeddings, where the weights are tuned in the process of training the downstream model, allowing the downstream model to give more weight to layers that are more relevant to valence prediction. The downstream model then makes a first linear projection to a fixed size of 256, before taking an unweighted average over the audio sequence, and then a second linear projection down to a single number. We train the downstream model using mean squared error loss, Adam optimization (Kingma and Ba, 2014), and for a maximum of 20,000 steps. These hyperparameters, as well as the model's architecture, are based on those used for the emotion recognition task by Yang et al. (2021), although we decrease the total number of steps, as our valence datasets are significantly smaller than those used originally, and switch to an MSE loss, as we are predicting continuous ratings of valence rather than categorical emotions. We train three downstream models for each pre-trained model, one using a learning rate of $10^{-3}$, one using a learn-

ing rate of $10^{-4}$, and one using a learning rate of $10^{-5}$. We then use predictions from all downstream models when calculating Cohen's $d$.

## G Embedding Association Tests for U.K. English

To test the extent to which our results extend beyond U.S. English, we also perform tests using U.K. English.

### G.1 Stimuli

We now describe the stimuli used for tests concerning U.K. English.

#### G.1.1 Target Stimuli

**British and Foreign Accented Speakers** We adapt the audio IAT performed in (Romero-Rivas et al., 2021) comparing speech from a native speaker of British Received (BR) English with speech from a speaker of Spanish accented English. The original test uses speech samples read by two female actors, one using BR English, and the other using Spanish accented English. The actors each read 16 neutrally-valenced phrases, for example "table." Some participants in the original IAT receive versions of the audio stimuli with background noise added, while others received stimuli free of background noise, giving two versions of each sample. We therefore test 32 samples for each group.

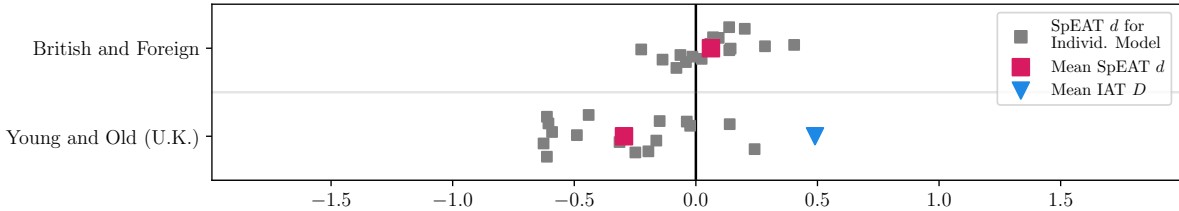

Figure 6: SpEAT $d$ values showing that results do not appear to generalize to U.K.-accented speech. The overall mean IAT $D$ value for the British and Foreign test was not provided by Romero-Rivas et al. (2021), however they calculate a $D$ value related to British accents of 0.60 and a $D$ value related to Foreign accents of -0.46.

**Young and Old Speakers** We also perform a test of age for U.K. English, and use a similar target dataset as for U.S. English, sourced from the SAA (Weinberger, 2015). There are 82 speech samples in the dataset from speakers born in the United Kingdom. A tradeoff needed to be made between larger group sizes, which can provide more precision in tests, and smaller group sizes, which provide better representations of the categories of young and old. The value of 12 is within typical ranges for sample sizes of past EATs. The young category consists of speakers between ages 18 and 21 while speakers in the old category are between ages 38 and 72. As in the U.S. dataset, we balance our sample on gender.

### G.1.2 Attribute Stimuli: EU-Emotion Stimulus Set

When performing the SpEAT using stimuli based on U.K. accents, we use EU-Emotion Stimulus Set (EUESS) rather than the MESS to represent positive and negative valence (O'Reilly et al., 2016; O'Reilly et al., 2012). The EUESS is a corpus of 695 recordings made by actors and rated by listeners in the United Kingdom. Actors were speakers of British English, and were asked to read sets of sentences each portraying one of twenty categorical emotional states. The corpus contains both 255 semantically emotional sentences and 440 semantically neutral sentences, of which we only use the semantically neutral sentences, in order to center our work on acoustics. (Semantically neutral sentences were those that were possible to read in multiple emotional states, for example "I knew it would happen.") A minimum of 20 listeners provided ratings of valence for each recording. Actors varied in age and gender, and so we again sample positive and negative valence within speaker to minimize the influence of speaker-based confounders. We take three recordings from each speaker for

each of the valence poles, giving us 54 recordings for positive valence and 54 recordings for negative valence. To establish that any potential differences in association between gender and valence measured by the SpEAT are not due to how the samples were rated by humans, we ensure that human ratings of valence do not differ based on the speaker's gender. We perform a Welch's unequal variances $t$-test to evaluate whether the ratings of valence differ by gender on average. A $t$-test is appropriate due to the sample size of the dataset, and we use a Welch's $t$-test rather than a Student's $t$-test due to its robustness (Delacre et al., 2017; West, 2021). We do not find evidence that clips from male and female speakers are rated differently on valence on average ($t(437) = 0.81$, $p = 0.42$), implying that any differences in association with valence between gender are not due to differences in how the audio clips were rated by humans. We also establish that potential differences in association between age and valence measured by the SpEAT are not due to how the samples were rated by humans. We test whether age has a bivariate relationship with valence using a simple linear regression model fit using ordinary least squares. A regression is appropriate due to the continuous nature of both variables. We do not find evidence for a relationship between speaker age and valence ($t(438) = -0.90$, $p = 0.37$), implying that differences in association with valence by age are not due to differences in how the audio clips were rated by humans. Analysis does not indicate any departures from the linear regression model, implying that it is not inappropriate for the data. A residual plot for the model between age and valence in the EUESS is shown in Figure 7. Based on the lack of pattern in the local average line (in red), we do not see evidence for a nonlinear relationship between the independent and dependent variables. We also do not see evidence for heteroscedasticity in the

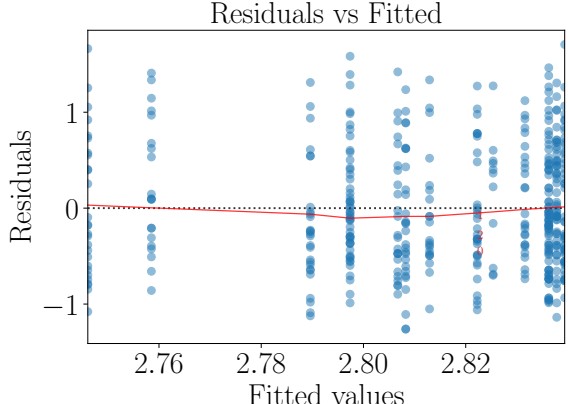

Figure 7: Residual plot for regression between age and valence in EUESS dataset, which contains speech from speakers of U.K. English rated on valence. We do not see evidence for heteroscedasticity or for non-linearity, suggesting that assumptions from the regression model are not violated.

residuals. Due to the sample size (440), an assumption of normality is not necessary for use of the $t$-test, and we therefore do not perform a test for normality of the residuals.

## G.2 Test Results

### G.2.1 British and Foreign Accented Speakers

10 of 16 models show a positive SpEAT $d$ value. We note the British and Foreign Accented samples are based on only 16 unique recordings from each of the two unique speakers, and for this reason may not be fully representative of the intended concepts. The locality of the speech (being from a British speaker) may also affect the results.

### G.2.2 Young and Old Speakers

Only two models show bias against speech from older speakers from the United Kingdom. We note that due to the data available in the SAA, the U.K. test only involves twelve speech samples to represent each concept. We also note that the speech samples used to represent older speakers in the U.K. also have a much lower age cutoff than older speakers in the United States, which may mean the target concepts are not as differentiated from each other.