# OpenReview forum: "Pre-trained Speech Processing Models Contain Human-Like Biases that Propagate to Speech Emotion Recognition"
_EMNLP/2023/Conference — EMNLP 2023 Findings_

### Official Review · Reviewer_PR28 · 2023-07-21

**Soundness:** 4

**Excitement:**

4: Strong: This paper deepens the understanding of some phenomenon or lowers the barriers to an existing research direction.

**Paper Topic And Main Contributions:**

The paper proposes a test (SpEAT) to measure bias for/against certain demographics in pretrained speech models such as WavLM. SpEAT is based on comparing embeddings of speech by the demographic groups in question to embeddings of speech that is deemed of low and high valence, respectively. The results show that oftentimes, the models actually show bias comparable to bias reported for humans. Moreover, the authors provide evidence that these biases are often propagated to the downstream taks of speech emotion recognition.

**Questions For The Authors:**

A: how many seeds have been used to train the downstream models?

**Reasons To Accept:**

- The paper addresses a relevant research question
- It provides a notably large number of experiments which cover different pretrained models and different kinds of biases
- The reported experiments seem to be designed very carefully (selection of speech samples etc.)
- The paper reads well and is well-structured

**Reasons To Reject:**

(EDIT: the questions have been sufficiently addressed in the authors' response)

- The paper does not consider that such models have been found to also pick up linguistic knowledge similar to large Language Models (Wagner, Johannes, et al. "Dawn of the transformer era in speech emotion recognition: closing the valence gap." IEEE Transactions on Pattern Analysis and Machine Intelligence (2023).). This may be partly relevant here, as (if I understand correctly) the speakers in the European-American vs. African-American dataset do not all say the same text, but deliver individual answers to interview questions.

- It is not clarified if several seeds have been used for the fine-tuning process. The paper reads like there was only one seed, which would be a bit of a limitation, as oftentimes, finetuning results are heavily seed-dependent.

**Reproducibility:**

4: Could mostly reproduce the results, but there may be some variation because of sample variance or minor variations in their interpretation of the protocol or method.

**Reviewer Confidence:**

3: Pretty sure, but there's a chance I missed something. Although I have a good feel for this area in general, I did not carefully check the paper's details, e.g., the math, experimental design, or novelty.

**Typos Grammar Style And Presentation Improvements:**

- The Ethics Statement (9) sounds more like a Discussion/Conclusion in large parts, e.g. when suggesting approaches to tackle biases. I suggest to put most of these considerations into the main part of the paper in case of acceptance.

- Equations (line 257, line 269) should be put in a equation environment in LaTex.

- 261-262: clarifying that A represents pleasantness and B unpleasantness in this example would help
- 343: appears -> appear
- 589: wording a bit unclear here: "can easily learn" - it is clear that these models *can* learn such biases, the question here seems rather whether the pretrained models *did in fact* adopt human biases during pretraining?

---

> ### Author Rebuttal · Authors · 2023-08-29
>
> Dear Reviewer PR28,
>
>
> Thank you for your comments and questions. We have addressed these in line below.
>
>
> ***
>
>
> > The paper does not consider that such models have been found to also pick up linguistic knowledge similar to large Language Models (Wagner, Johannes, et al. "Dawn of the transformer era in speech emotion recognition: closing the valence gap." IEEE Transactions on Pattern Analysis and Machine Intelligence (2023).). This may be partly relevant here, as (if I understand correctly) the speakers in the European-American vs. African-American dataset do not all say the same text, but deliver individual answers to interview questions.
>
> Because embeddings for speech from African-American speakers tend to be closer to negative valence in our tests of pre-trained models (and tend to be predicted as lower in valence by downstream models) we think that if linguistic differences were to explain the results in the AA/EA tests, the linguistic content of the speech sourced from African-American speakers would need to be lower in valence than the linguistic content of the speech sourced from European-American speakers. To evaluate this, we passed the transcripts of the speech from the AA/EA test through five popular sentiment analysis tools [1-5].
>
> In all but one tool we used, the transcripts from African-American speakers in fact have *higher average positive sentiment* and *lower average negative sentiment* than transcripts from European-American speakers. The other tool (VADER) showed the transcripts from African-American speakers as higher in *both* average positive sentiment and average negative sentiment, but outputs a compound score combining positive, negative, and neutral sentiment, which was higher for the transcripts of African-American speakers.
>
> Regardless of these results, however, we think it is worth noting that in all other tests the content of the speech is matched across demographic groups.
>
> ***
> > It is not clarified if several seeds have been used for the fine-tuning process. The paper reads like there was only one seed, which would be a bit of a limitation, as oftentimes, finetuning results are heavily seed-dependent.
>
> For the results reported in the paper, we do in fact only use one seed when training downstream models to predict valence. In early experiments, however, we performed light hyperparameter tuning by training downstream models using the learning rates 1e-3, 1e-4, and 1e-5. We only discuss the models fit using the 1e-4 learning rate in the paper, in order to follow Yang et al.’s [6] method for adapting pre-trained models to downstream tasks as closely as possible. We think this question raises a good point regarding the reliability of the results when only 16 downstream models are used, however, and for this reason chose to report on the other models below. For the most part, we do not see large performance disparities between the models fit with different learning rates. Excluding one downstream model, which was fit using a learning rate of 1e-5 on embeddings from the Libri-Light version of wav2vec 2.0 Large and had a negative $R^2$, the average $R^2$ values for the 1e-3, 1e-4 and 1e-5 models were 0.91, 0.88, 0.89, respectively. We will plan on switching to discussing results from all three models in our paper, if it is accepted.
>
> When considering the additional models, we do not find that our results change significantly. In the paper, we report that when using only downstream models trained with the 1e-4 learning rate, in “65 of the 96 (68%) tests performed, the group more associated with positive valence by the pre-trained model is also more associated with positive valence by the downstream SER model.” Similarly, when using predictions from all three downstream models, there are 66 of 96 (69%) of tests where the group more associated with positive valence by the pre-trained model is also more associated with positive valence by the downstream SER models.
>
> In the SER experiments we measured bias in the downstream task using Cohen’s $d$, an effect size metric summarizing the difference in mean predicted valence between the two social groups. We do not find that including the predictions from the models trained with other learning rates results in Cohen’s $d$ values that we see as substantially different. When we compare predictions from all three downstream models to predictions from only the 1e-4 model, we find that the average absolute difference in Cohen’s $d$ is 0.13. This is relative to an average Cohen’s $d$ magnitude of 0.79 when using the 1e-4 models or 0.71 when using the 1e-3, 1e-4, and 1e-5 models together. In the paper we state that “in all tests excluding that related to human and synthesized speech, the average Cohen’s $d$ across all 16 speech models is in the same direction as the average SpEAT $d$ across the models.“ This does not change when including the additional downstream models.
>
> ***
>
> Thank you for your suggestions and concerns, including those raised regarding style. We agree that a portion of the ethics statement (namely, the second paragraph) would be better placed in the discussion section, and will plan on making this change, along with the others you suggest, if our work is accepted.
>
> ***
>
> [1] Daniel Loureiro, Francesco Barbieri, Leonardo Neves, Luis Espinosa Anke, and Jose Camacho-collados. 2022. TimeLMs: Diachronic Language Models from Twitter. In Proceedings of the 60th Annual Meeting of the Association for Computational Linguistics: System Demonstrations, pages 251–260, Dublin, Ireland. Association for Computational Linguistics.
>
> [2] Francesco Barbieri, Luis Espinosa Anke, and Jose Camacho-Collados. 2022. XLM-T: Multilingual Language Models in Twitter for Sentiment Analysis and Beyond. In Proceedings of the Thirteenth Language Resources and Evaluation Conference, pages 258–266, Marseille, France. European Language Resources Association.
>
> [3] Francesco Barbieri, Jose Camacho-Collados, Luis Espinosa Anke, and Leonardo Neves. 2020. TweetEval: Unified Benchmark and Comparative Evaluation for Tweet Classification. In Findings of the Association for Computational Linguistics: EMNLP 2020, pages 1644–1650, Online. Association for Computational Linguistics.
>
> [4] [Seethal/sentiment_analysis_generic_dataset · Hugging Face](https://huggingface.co/Seethal/sentiment_analysis_generic_dataset)
>
> [5] Hutto, C., & Gilbert, E. (2014). VADER: A Parsimonious Rule-Based Model for Sentiment Analysis of Social Media Text. Proceedings of the International AAAI Conference on Web and Social Media, 8(1), 216–225. https://doi.org/10.1609/icwsm.v8i1.14550
>
> [6] Yang, S., Chi, P.-H., Chuang, Y.-S., Lai, C.-I. J., Lakhotia, K., Lin, Y. Y., Liu, A. T., Shi, J., Chang, X., Lin, G.-T., Huang, T.-H., Tseng, W.-C., Lee, K., Liu, D.-R., Huang, Z., Dong, S., Li, S.-W., Watanabe, S., Mohamed, A., & Lee, H. (2021). SUPERB: Speech Processing Universal PERformance Benchmark. Proc. Interspeech 2021, 1194–1198. https://doi.org/10.21437/Interspeech.2021-1775

---

### Official Review · Reviewer_FQKD · 2023-07-31

**Soundness:** 4

**Excitement:**

4: Strong: This paper deepens the understanding of some phenomenon or lowers the barriers to an existing research direction.

**Missing References:**

The Paper "GigaSpeech: An Evolving, Multi-domain ASR Corpus with 10,000 Hours of Transcribed Audio" was published in  interspeech 2021 (10.21437/Interspeech.2021-1965). It's better to cite that instead of arXiv

The citation "202" and "2023c" in page 14 point to the same reference, probably an error


**Paper Topic And Main Contributions:**

The paper introduces a novel method to identify biases (disability, race, gender, accent, age) in pre-trained speech models. Experiments are presented including popular speech models: wav2vec 2.0, HuBERT, WavLM, Whisper. The paper also proves that the pre-trained speech models learn the biases available in the large training dataset and can propagate those to the downstream task of Speech Emotion Recognition (SER).


**Questions For The Authors:**

   If you using public models, it would be useful to disclose the actual models you used. For example, could you please provide the exact links or identifiers for the wav2vec2.0 models you used? This would greatly enhance the transparency of your study and make it easier for readers to understand your methodology and reproduce your results. Additionally, as certain models like Whisper Large have version 1 and an updated version 2, providing the specific versions used can help avoid confusion.

  Do you think the lack of synthesized speech in pre-training models could explain the difference between upstream and downstream models for synthesized speech (as shown in figure 2). Are you considering experimenting with models pre-trained on a significant synthesized speech data? Might this change the bias behaviors depicted in figures 1 and 2. Furthermore, could using synthesizing speech data (based on actual speech) to minimize biases in pre-training models create a significant bias between synthesized and human speech, while reducing other biases? I'd appreciate your views on this.


**Reasons To Accept:**

    The paper introduces a new method called Speech Embedding Association Test (speEAT) for detecting biases in pre-trained models. This is a noteworthy contribution as the method can be applied to a broad range of widely-adopted model architectures. Moreover, it is important to note that this research addresses an under-explored area in the field.

    The results of this paper also raise awareness about the social biases within speech models. While the paper primarily focuses on SER, it also raises ethical discussion about the necessity of developing unbiased pre-trained speech models in the future.

    The code for SpeEAT method will also be published, which could server as a good foundation to develope more sophisticated bias detection method in the future.


**Reasons To Reject:**

It did not become completely clear to me how much the SpEAT adds on the previously proposed EAT methods.

The statistical significance of the results are somewhat difficult to assess.


**Reproducibility:**

4: Could mostly reproduce the results, but there may be some variation because of sample variance or minor variations in their interpretation of the protocol or method.

**Reviewer Confidence:**

2: Willing to defend my evaluation, but it is fairly likely that I missed some details, didn't understand some central points, or can't be sure about the novelty of the work.

**Typos Grammar Style And Presentation Improvements:**

There are a few very long sentences: E.g. On page 5: "Stimuli are matched on gender as well as approximate age, with the exception of the female and male test, which is not matched on gender, the young and old test, which is not matched on age, the human and synthesized test, which is not matched on age, 6 and the abled and disabled test (for which age information was not made available for speakers)."

On page 7: There might be a missing word in this sentences: "Work in computer vision has suggested that biases detected by EATs may arise from co-occurrence in the training data (Steed and Caliskan, 2021), and we hypothesize that co-occurrence may contribute here as well, for example if speakers who use a ??? speech style tends to more frequently express positive valence than others in the training set."

On page 14 (two "a"): embeds each of the chunks in latent space using a a set of feature encoding layers

On page 19 (two "are"): There are are 82 speech samples in the dataset

Worfe et al., 2023 reference should at least capitalize the word "AI" (Contrastive Language-Vision {AI} Models Pretrained on Web-Scraped Multimodal Data Exhibit Sexual Objectification Bias)

There are a few more errors in capitalize the proper noun in reference (like english instead of English), please check that.

---

> ### Author Rebuttal · Authors · 2023-08-29
>
> Dear Reviewer FQKD,
>
>
> Thank you for your comments and suggestions. We provide in line responses below.
>
>
> ***
>
>
> > It did not become completely clear to me how much the SpEAT adds on the previously proposed EAT methods.
>
>
> The aim of our work was to study bias in speech processing models, and we chose to do so using the EAT, a method that has been studied extensively in the Natural Language Processing and Computer Vision communities. We saw making methodological improvements to EATs as mostly out of scope. Once embeddings have been extracted from a speech model, the SpEAT in fact does not differ in process from a form of the Sentence Embedding Association Test. We see our primary additions to the EAT literature as 1) introducing an EAT for models in a new modality, 2) showing that EATs generalize across modalities and architectures, and 3) adding evidence that biases found with EATs can propagate to downstream tasks.
>
>
> We note one methodological contribution, however, which is that we have shown how bootstrapping can be used to estimate statistical properties of the EAT, such as the Standard Error. We hope that this can be built upon by others in the future, for example for constructing confidence intervals for EAT $d$ values, or for estimating the sample size that would be necessary to achieve a confidence interval of a certain size.
>
>
> ***
>
>
> >The statistical significance of the results are somewhat difficult to assess.
>
>
> In light of this comment, we ran hypothesis tests for each of the individual SpEATs performed. Prior works concerning EATs have typically touched on statistical significance using a permutation-based Null Hypothesis Significance Test (NHST) [1,2,3]. This NHST tests the null hypothesis that the EAT $d$ in a specific model is equal to 0 for the specific set of concepts against the alternative hypothesis that the EAT $d$ is larger than 0.
>
>
> We perform this test for each of the 6 sets of concepts in each of the 16 speech models, and find that 63 of 96 tests (66%) are significant at the 0.01 level. Of the tests significant at the 0.01 level, 46 (73%) are also significant after a Bonferroni correction for multiple comparisons. We will include these results in the appendix of our work, if it is accepted. We are also happy to answer further questions concerning these tests.
>
>
> ***
>
>
> > If you using public models, it would be useful to disclose the actual models you used. For example, could you please provide the exact links or identifiers for the wav2vec2.0 models you used? This would greatly enhance the transparency of your study and make it easier for readers to understand your methodology and reproduce your results. Additionally, as certain models like Whisper Large have version 1 and an updated version 2, providing the specific versions used can help avoid confusion.
>
>
> We apologize for the lack of detail. For the Whisper family we use the 1.0 versions of the models, and for the wav2vec 2.0 model family we use the non fine-tuned versions of the models. For the wav2vec2.0 family, the model versions are provided on lines 1228-1233 in the appendix of our submission. For the HuBERT family, the versions are provided on lines 1273-1275. For the WavLM family, the versions are provided on lines 1328-1329. Finally, for the Whisper family, the versions are provided on lines 1355-1358.
>
>
> We will add the missing information to future versions of our work, which we believe should be sufficient for finding the models that we used. In order to facilitate replication, however, we have also included links to the wav2vec 2.0, WavLM, and HuBERT models used below. We will include these links in the main README for the code released with the project as well, if our work is accepted. We have not added links for the Whisper models, as these are downloaded automatically by the scripts that we have provided.
>
>
> * [wav2vec 2.0 Base (No finetuning/Librispeech)](https://github.com/facebookresearch/fairseq/blob/4db264940f281a6f47558d17387b1455d4abd8d9/examples/wav2vec/README.md)
> * [wav2vec 2.0 Large (No finetuning/Librispeech)](https://github.com/facebookresearch/fairseq/blob/4db264940f281a6f47558d17387b1455d4abd8d9/examples/wav2vec/README.md)
> * [wav2vec 2.0 Large (No finetuning/Libri-Light)](https://github.com/facebookresearch/fairseq/blob/4db264940f281a6f47558d17387b1455d4abd8d9/examples/wav2vec/README.md)
> * [HuBERT Base (No finetuning/Librispeech)](https://github.com/facebookresearch/fairseq/tree/4db264940f281a6f47558d17387b1455d4abd8d9/examples/hubert)
> * [HuBERT Large (No finetuning/Libri-Light)](https://github.com/facebookresearch/fairseq/tree/4db264940f281a6f47558d17387b1455d4abd8d9/examples/hubert)
> * [HuBERT Extra Large (No finetuning/Libri-Light)](https://github.com/facebookresearch/fairseq/tree/4db264940f281a6f47558d17387b1455d4abd8d9/examples/hubert)
> * [WavLM Base](https://github.com/microsoft/unilm/tree/f4695ed0244a275201fff00bee495f76670fbe70/wavlm)
> * [WavLM Base+](https://github.com/microsoft/unilm/tree/f4695ed0244a275201fff00bee495f76670fbe70/wavlm)
> * [WavLM Large](https://github.com/microsoft/unilm/tree/f4695ed0244a275201fff00bee495f76670fbe70/wavlm)
>
>
> ***
>
>
> > Do you think the lack of synthesized speech in pre-training models could explain the difference between upstream and downstream models for synthesized speech (as shown in figure 2). Are you considering experimenting with models pre-trained on a significant synthesized speech data? Might this change the bias behaviors depicted in figures 1 and 2. Furthermore, could using synthesizing speech data (based on actual speech) to minimize biases in pre-training models create a significant bias between synthesized and human speech, while reducing other biases? I'd appreciate your views on this.
>
>
> The lack of synthesized speech in the pre-training datasets used by these speech models may explain some results in the experiments comparing human and synthesized speech. It’s difficult to say how synthetic data may change biases, however, or how biases relate to one another, especially without better information on the dialectal and demographic characteristics of the speech that make up common speech datasets. The quality of synthesized speech has been improving though, and if newer speech processing models are trained with synthetic speech, we may see interesting developments in the future.
>
>
> ***
>
>
> Thank you for raising the concerns above, as well for the grammar and style suggestions. We will address these in the final manuscript if the paper is accepted, and will also make another pass at reviewing the paper ourselves.
>
>
> ***
>
> [1] May, C., Wang, A., Bordia, S., Bowman, S. R., & Rudinger, R. (2019). On Measuring Social Biases in Sentence Encoders. Proceedings of the 2019 Conference of the North American Chapter of the Association for Computational Linguistics: Human Language Technologies, Volume 1 (Long and Short Papers), 622–628. https://doi.org/10.18653/v1/N19-1063
>
> [2] Steed, R., & Caliskan, A. (2021). Image Representations Learned With Unsupervised Pre-Training Contain Human-like Biases. Proceedings of the 2021 ACM Conference on Fairness, Accountability, and Transparency, 701–713. https://doi.org/10.1145/3442188.3445932
>
> [3] Caliskan, A., Bryson, J. J., & Narayanan, A. (2017). Semantics derived automatically from language corpora contain human-like biases. Science, 356(6334), 183–186. https://doi.org/10.1126/science.aal4230

---

### Official Review · Reviewer_QLo4 · 2023-08-10

**Typos Grammar Style And Presentation Improvements:** 105
**Soundness:** 3

**Excitement:**

4: Strong: This paper deepens the understanding of some phenomenon or lowers the barriers to an existing research direction.

**Paper Topic And Main Contributions:**

The article describes a new method for detecting bias in pre-trained speech models, demonstrates using this method that many existing state-of-the-art models exhibit bias, and demonstrates that downstream systems relying on such models' output also exhibit bias.

**Reasons To Accept:**

The treatment of bias for speech models is novel.

**Reasons To Reject:**

The treatment of bias is likely exotic for most readers. The current manuscript misses some opportunities for making the related concepts plain and crisp. The article would likely be much improved if it: (1) provided a single, highlighted definition of the (authors') operationalized definition of bias; (2) included a conceptual description of how that bias is computed; (3) argued more cogently for the relationship between "valence" (allegedly descriptive of how a person feels) and "pleasantness" (supposedly descriptive of how a person makes another person feel); and (4) argued more cogently for the relationship between "valence" in a downstream SER system and "bias" as it is found in earlier portions of this article. Several of these concepts/attempts are provided loosely, scattered throughout the article, or made/hinted at only in parentheses.

Lines 596 and 606 mention "(speaking) style" -- this appears late in the article, is not defined, and thereby retains an unclear relationship with the authors' arguments.

**Reproducibility:**

3: Could reproduce the results with some difficulty. The settings of parameters are underspecified or subjectively determined; the training/evaluation data are not widely available.

**Reviewer Confidence:**

3: Pretty sure, but there's a chance I missed something. Although I have a good feel for this area in general, I did not carefully check the paper's details, e.g., the math, experimental design, or novelty.

---

> ### Author Rebuttal · Authors · 2023-08-29
>
> Dear Reviewer QLo4,
>
>
> Thank you for your review. We have tried to clarify the concepts and arguments that you raised concerns about here, and if our work is accepted we will use some of the additional space for the same purpose.
>
>
> ***
>
>
> > (3) argued more cogently for the relationship between "valence" (allegedly descriptive of how a person feels) and "pleasantness" (supposedly descriptive of how a person makes another person feel)
>
>
> We apologize for not defining valence more clearly in our work’s introduction. Valence is a concept used for categorizing emotions, and is in fact often defined directly in psychology literature as “pleasantness” or “pleasure,” or used interchangeably with these terms [4-6]. Valence is a part of many dimensional models of emotion, which consider emotions to be composed of different (sometimes independent) dimensions. For example, the circumplex model [4] categorizes emotions using valence and arousal. This model considers excitement to have positive valence and positive arousal, contentment to have positive valence and negative arousal, distress to have negative valence and positive arousal, and so on. We have used the terms “valence” and “pleasantness” interchangeably, and apologize for the confusion this has caused. If our work is accepted we will plan on both defining valence more clearly at the outset and removing references to “pleasantness” that are not used for defining valence.
>
>
> ***
>
>
> > (2) included a conceptual description of how that bias is computed
>
>
> The SpEAT considers where in a model’s embedding space two concepts related to demographics, such as European-American and African-American, are located, relative to the concepts of positive and negative valence. A SpEAT involves passing stimuli representing these four concepts into a speech model, and retrieving the respective embeddings. The SpEAT is computed by first calculating how close embeddings representing one demographic concept are to positive valence relative to negative valence, then comparing this to how close embeddings representing the other demographic concept are to positive and negative valence. The SpEAT gives an effect size $d$, which when large in magnitude indicates that one group tends to be closer than the other group to positive valence relative to negative valence. A positive SpEAT $d$ indicates that the group that is closer to positive valence is also the group that tends to be more associated with positive valence by humans.
>
>
> ***
>
>
> > (1) provided a single, highlighted definition of the (authors') operationalized definition of bias
>
>
> Bias is a complex concept with many, often contradictory, ways of being operationalized [2], and we apologize for not describing how the SpEAT relates to bias more clearly. We consider a large SpEAT $d$ magnitude to be a bias, although it is not the only bias we consider in the paper. The other biases we consider are bias in the downstream task of Speech Emotion Recognition (SER), and bias in humans measured using Implicit Association Tests (IATs). In the downstream task of SER, we consider a disparity in predicted valence between matched speech samples coming from two demographic groups to be a bias, (which we will discuss in more detail in the following in line comment). In humans we consider a large IAT $D$ magnitude to be a bias.
>
>
> We view Embedding Association Tests (EATs) as measurements of bias in fact because of the relationship between distances in embedding spaces and other types of bias. Distance between embeddings has been found to be associated with biases in society, both in our work as well as in related work in Natural Language Processing and Computer Vision. In related work, it has been found that distances between embeddings for models trained on text from different time periods reflect social conditions from those time periods [3], and that distances between embeddings reflect real world statistics, like the proportion of people holding a certain job who are male versus female [1]. In our work, we find that SpEAT $d$ values tend to be in the same direction as average IAT $D$ values. We also find that the SpEAT is associated with disparities in predicted valence in downstream SER models.
>
>
> ***
>
>
> > (4) argued more cogently for the relationship between "valence" in a downstream SER system and "bias" as it is found in earlier portions of this article
>
>
> As noted above, we consider a disparity in predicted valence between speech samples coming from two demographic groups to be a bias in the downstream task of SER. The focus of this portion of our work is on how biases found in pre-trained models with the SpEAT may propagate to downstream tasks. To study this, we train downstream models (one for each pre-trained speech processing model) to predict the valence of speech samples, a form of SER. We then use these models to predict the valence of speech samples coming from people with the demographic characteristics that we studied in the pre-trained models. For example, we predict the valence of speech coming from African-American speakers and European-American speakers.
>
>
> When testing for bias, the samples that we predict the valence of consist either of speech read in a controlled environment or extemporaneous speech taken from socio-linguistic interviews. In addition to being collected in similar fashions, the clips were also often matched on gender and approximate age, as we describe in Section 4 of the paper. We consider disparities in predicted valence to be a bias because of the matched nature between the clips across the demographic groups studied.
>
>
> ***
>
>
> In addition to the requested clarifications, which we think will improve the readability of the paper, we’d also like to thank you for your attention to detail regarding grammar. We will review the paper again for grammatical errors if it is accepted.
>
>
> ***
>
>
> [1] Caliskan, A., Bryson, J. J., & Narayanan, A. (2017). Semantics derived automatically from language corpora contain human-like biases. Science, 356(6334), 183–186. https://doi.org/10.1126/science.aal4230
>
>
> [2] Blodgett, S. L., Barocas, S., Daumé III, H., & Wallach, H. (2020). Language (Technology) is Power: A Critical Survey of “Bias” in NLP. Proceedings of the 58th Annual Meeting of the Association for Computational Linguistics, 5454–5476. https://doi.org/10.18653/v1/2020.acl-main.485
>
>
> [3] Garg, N., Schiebinger, L., Jurafsky, D., & Zou, J. (2018). Word embeddings quantify 100 years of gender and ethnic stereotypes. Proceedings of the National Academy of Sciences of the United States of America, 115(16), E3635–E3644. https://doi.org/10.1073/pnas.1720347115
>
>
> [4] Russell, J. A. (1980). A circumplex model of affect. Journal of Personality and Social Psychology, 39(6), 1161–1178. https://doi.org/10.1037/h0077714
>
>
> [5] Nielen, M. M. A., Heslenfeld, D. J., Heinen, K., van Strien, J. W., Witter, M. P., Jonker, C., & Veltman, D. J. (2009). Distinct brain systems underlie the processing of valence and arousal of affective pictures. Brain and Cognition, 71(3), 387–396. https://doi.org/https://doi.org/10.1016/j.bandc.2009.05.007
>
>
> [6] Morgan, S. D. (2019). Categorical and Dimensional Ratings of Emotional Speech: Behavioral Findings From the Morgan Emotional Speech Set. Journal of Speech, Language, and Hearing Research, 62(11), 4015–4029. https://doi.org/10.1044/2019_JSLHR-S-19-0144

---

### Official Review · Reviewer_b7KZ · 2023-08-12

**Soundness:** 3

**Excitement:**

4: Strong: This paper deepens the understanding of some phenomenon or lowers the barriers to an existing research direction.

**Paper Topic And Main Contributions:**

This paper addresses the question - do large pre-trained speech models carry markers of bias similar to humans? The authors extend the technique of Embedding Association Tests (EAT) to speech forming SpEAT to first (a) validate whether prediction of valence (used for Affect prediction or SER) differs across 6 social groups, and (b) finally examine the impact of this bias on Speech Emotion Recognition.


Prior work in diagnosing bias from speech models has focused on reporting performance differences across data representing different social groups, and in reporting downstream performance of fine-tuned large-scale pre-trained speech models on these different groups. This work in contrast uses pre-trained model embeddings directly and compares them to human biases obtained through IATs. The authors consider activations from multiple layers in the large-scale model, compute temporal averages within every layer, and then sum across layers to obtain the embedding they test with.


The authors verify (a) whether the pretrained speech models carry similar biases to humans by comparing SpEAT d values to IAT test D values from previous work, (b) whether the biases propagate when fine-tuning for downstream Speech Emotion Recognition, and (c) how much SpEAT values vary when changing the sample size of the stimuli (participants ). The authors find that many of the models carry biases similar to humans, and that biases propagate to downstream SER models.

**Questions For The Authors:**

A. Do the authors have intuitions on why model predictions seem to have similar biases to humans? I would have thought that some of the bias may be because of labelling by annotators, but it appears that models pre-trained in a self-supervised manner (without any labels) carry this bias as well.

B. Could you please clarify why summing the activations across multiple layers is useful/necessary when works like Chen et. al. and Pasad et. al. [1] have identified methods to estimate layers that may correlate with valence or emotion?

C. There are multiple categories of large-scale speech models today - did the authors find any commonalities in the type and extent of bias in models (a) trained on the same data or (b) trained in the same manner? Some conclusions in this respect would be useful for future work.

[1] Comparative layer-wise analysis of self-supervised speech models, Ankita Pasad, Bowen Shi and Karen Livescu, ICASSP 2023

**Reasons To Accept:**

A. This paper focuses on biases of pre-trained models - an important topic for the speech community to study and actively pursue as large-scale models are being used and deployed in real-world scenarios.

B. This paper compares speech model biases to human biases which is interesting and novel in application if not methodology to the best of my knowledge.

C. The paper is detailed in describing experimental goals, experiments, and results and tests a broad variety of speech models including Wav2Vec variants, Whisper variants, and HUBERT variants.

**Reasons To Reject:**

A. The method used to generate embeddings, i.e., temporal averaging per layer and summing across layers is quite possibly error-ridden. As the authors have pointed out, different layers of large-scale pre-trained models are more useful to particular tasks, and summing activations across different layers may result in less discriminative features.

B. Further, SpEAT evaluations appear to be done with relatively small sample sizes, which is likely suboptimal. In many cases, this appears to be due to the availability of data, but this remains a challenge unaddressed.

EDIT:

The authors have provided a convincing justification that addresses part B.

**Reproducibility:**

3: Could reproduce the results with some difficulty. The settings of parameters are underspecified or subjectively determined; the training/evaluation data are not widely available.

**Reviewer Confidence:**

3: Pretty sure, but there's a chance I missed something. Although I have a good feel for this area in general, I did not carefully check the paper's details, e.g., the math, experimental design, or novelty.

**Typos Grammar Style And Presentation Improvements:**

The introduction is highly technical and hard to read for a person unfamiliar with EATs - at a high level, I suggest maybe going through it and simplifying it where possible for non-expert readers.

---

> ### Author Rebuttal · Authors · 2023-08-29
>
> Dear Reviewer b7KZ,
>
> Thank you for your feedback. We have included in line responses to your comments and questions below.
>
> ***
>
>
> > A. The method used to generate embeddings, i.e., temporal averaging per layer and summing across layers is quite possibly error-ridden. As the authors have pointed out, different layers of large-scale pre-trained models are more useful to particular tasks, and summing activations across different layers may result in less discriminative features.
>
> While we note that the method we use for generating embeddings is taken from a prior EAT [1], we thought this concern might also be addressed by evaluating the extent to which our results would have changed had different aggregation strategies been used. Across layers, we tested aggregating by using either the sum, min, or max—or by simply selecting embeddings from the first layer, second layer, first quartile layer, median layer, third quartile layer, penultimate layer, or last layer. Across the temporal dimension, we aggregated either by taking the mean, min, or max.
>
> In the paper, we report that when using the sum (layers)/mean (temporal) to generate embeddings, 82% of SpEAT $d$s are positive. When considering all 30 aggregation strategies, we find that 69% percent of the 2,880 tests show positive SpEAT $d$s. In the paper we show that when using the sum/mean to generate embeddings, the SpEAT $d$ averaged across the 16 models is in the same direction as the related IAT $D$ from human experiments for 5 of the 6 social groups (83%) considered. We in fact find that considering all 30 aggregation strategies and 6 social groups, exactly 150 of 180 (83%) average SpEAT $d$ values are in the same direction as the related IAT $D$. We also find that there are more positive SpEAT $d$ values than negative when using all but one aggregation strategy. If our work is accepted, we will include these results in the final manuscript’s appendix. We are also happy to answer further questions concerning this experiment.
>
> ***
>
>
> > B. Further, SpEAT evaluations appear to be done with relatively small sample sizes, which is likely suboptimal. In many cases, this appears to be due to the availability of data, but this remains a challenge unaddressed.
>
> We use between 57 and 60 stimuli to represent each social group (or valence pole), resulting in between 234 and 240 stimuli being used in each test. These sample sizes are larger than those used in any test performed either by Caliskan et al. when introducing the Word Embedding Association Test [3] or by Steed and Caliskan when introducing the Image Embedding Association Test [2].
> Regardless of this context, however, we do not think that larger sample sizes are necessary to support the claims made in our work. This is because of the way that we are using the stimuli, which is to estimate, first, the SpEAT $d$ and, second, an effect size concerning the difference in predicted valence for speech coming from two groups of people.
>
> We see smaller sample sizes as suboptimal mainly in that they will result in less precise estimates of these statistics. In our work estimating the Standard Error (SE) of the SpEAT $d$, however, we found that across all tests and speech models, the average estimated SE was 0.14 when 50 samples were used to represent target concepts, and the maximum estimated SE for any condition was 0.20 when 50 samples were used, (with both mean and maximum SE monotonically decreasing as the number of samples increased beyond 50). We see these SEs as already fairly small relative to the size of the SpEAT $d$ values estimated (whose average magnitude was 0.70 across all sets of stimuli and models), and therefore do not think that, had the sample size been increased, the uncertainty associated with the SpEAT $d$ would have depreciated by a meaningful amount. We have not estimated SEs for Cohen's $d$, but do not anticipate the results to have been significantly different had we done so.
>
> With that said, we recognize that larger sample sizes may come along with a different method for generating stimuli (i.e., using speech samples from other sources), and that testing alternative methods of generating stimuli may indeed lead to interesting results. We think that using the SpEAT with stimuli from a variety of different sources may be a promising direction for future work.
>
>
> ***
>
>
> > A. Do the authors have intuitions on why model predictions seem to have similar biases to humans? I would have thought that some of the bias may be because of labelling by annotators, but it appears that models pre-trained in a self-supervised manner (without any labels) carry this bias as well.
>
> Work in other modalities [2] has suggested that co-occurrence may be related to EAT results. In the case of the speech data, this may occur if people of one social group tend to more frequently speak with positive valence than people of another social group. We think that co-occurrence in the pre-training data might be a good place to start if one were interested in investigating how these results arise.
>
> ***
> > C. There are multiple categories of large-scale speech models today - did the authors find any commonalities in the type and extent of bias in models (a) trained on the same data or (b) trained in the same manner? Some conclusions in this respect would be useful for future work.
>
> We did notice a slight model family effect, with models in the same family seeming to have somewhat of a tendency toward similar SpEAT $d$ values concerning a given social group. We performed a two-way ANOVA to confirm this, using the SpEAT $d$ value as the dependent variable and the social group and model family as the independent variables. The interaction between model family and social group was just below the typical p=0.05 significance threshold, (p=0.048), suggesting, as suspected, that there may be a slight effect of model family regarding a social group. The main effect of model family was not significant, however, suggesting that it is not the case that SpEAT $d$ values for a family tend to be located in the same area across all social groups. For example, Whisper SpEAT $d$ values tend to be similar to one another in the African-American and European-American test, but Whisper SpEAT $d$ values for the African-American and European-American test do not tend to be similar to Whisper SpEAT $d$ values from the Young and Old test. With that said, model family is related to both the way the models were trained and the training dataset used, and we don’t believe that the information we have available allows for strong conclusions regarding these factors.
>
> ***
>
> Thank you again for your time and effort. We will review the introduction for readability if our work is accepted.
>
> ***
> [1] May, C., Wang, A., Bordia, S., Bowman, S. R., & Rudinger, R. (2019). On Measuring Social Biases in Sentence Encoders. Proceedings of the 2019 Conference of the North American Chapter of the Association for Computational Linguistics: Human Language Technologies, Volume 1 (Long and Short Papers), 622–628. https://doi.org/10.18653/v1/N19-1063
>
> [2] Steed, R., & Caliskan, A. (2021). Image Representations Learned With Unsupervised Pre-Training Contain Human-like Biases. Proceedings of the 2021 ACM Conference on Fairness, Accountability, and Transparency, 701–713. https://doi.org/10.1145/3442188.3445932
>
> [3] Caliskan, A., Bryson, J. J., & Narayanan, A. (2017). Semantics derived automatically from language corpora contain human-like biases. Science, 356(6334), 183–186. https://doi.org/10.1126/science.aal4230

---

### Meta-Review · Area_Chair_BBzE · 2023-09-14

**Recommendation:** 4

**Metareview:**

The paper proposes a way to quantify an intrinsic bias in the pre-trained model and shows the observation of a human-like bias in the pre-trained model and downstream as well. The authors addressed several questions and concerns from reviewers and showed potential evidence to strengthen the paper upon final acceptance.

---

### Decision · Program_Chairs · 2023-10-07

**Decision:**

Accept-Findings

**Comment:**

The paper proposes a way to quantify an intrinsic bias in the pre-trained model and shows the observation of a human-like bias in the pre-trained model and downstream as well. The authors addressed several questions and concerns from reviewers and showed potential evidence to strengthen the paper upon final acceptance.